



# Gap-Free Global Annual Soil Moisture: 15km Grids for 1991-2018

Mario Guevara[1], Michela Taufer[2], Rodrigo Vargas[1]

[1]Department of Plant and Soil Sciences, University of Delaware, Newark, DE, United States.

[2]Department of Electrical Engineering and Computer Science, The University of Tennessee, Knoxville, TN, United States.

10   *Correspondence to:* Rodrigo Vargas (rvargas@udel.edu)



**Abstract.** Soil moisture is key for quantifying soil-atmosphere interactions. We provide a soil moisture pattern recognition framework to increase the spatial resolution and fill gaps of the ESA-CCI (European Space Agency-Climate Change Initiative v4.5) soil moisture dataset, which contains more than 40 years of satellite soil moisture global grids with a spatial resolution of ~27km. We use terrain parameters coupled with bioclimatic and soil type information to predict the finer-grained satellite soil moisture. We assess the impact of terrain parameters on the prediction accuracy by cross-validating the pattern recognition of soil moisture with and without the support of bioclimatic and soil type information. The outcome is a new dataset of gap-free global mean annual soil moisture and uncertainty for 28 years (1991-2018) across 15km grids. We use independent *in situ* records from the International Soil Moisture Network (ISMN, n=13376) and *in situ* precipitation records (n=4909) only for evaluating the new dataset. Cross-validated correlation between observed and predicted soil moisture values varies from r=0.69 to r=0.87 with root mean squared errors (RMSE, $m^3/m^3$) around 0.03 and 0.04. Our soil moisture predictions improve: (a) the correlation with the ISMN (when compared with the original ESA-CCI dataset) from r=0.30 (RMSE=0.09, ubRMSE=0.37) to r=0.66 (RMSE=0.05, ubRMSE=0.18); and (b) the correlation with local precipitation records across boreal (from r=<0.3 up r=0.49) or tropical areas (from r=<0.3 to r=0.46) which are currently poorly represented in the ISMN. Temporal trends show a decline of global annual soil moisture using: a) data from the ISMN (-1.5 [-1.8, -1.24]%, b) associated locations from the original ESA-CCI dataset (-0.87[-1.54, -0.17]%), c) associated locations from predictions based on terrain parameters (-0.85[-1.01, -0.49]%), and d) associated locations from predictions including bioclimatic and soil type information (-0.68[-0.91, -0.45]%). We provide a new soil moisture dataset that has no gaps and a finer resolution together with validation methods and a modeling approach that can be applied worldwide (Guevara, et al., 2020, https://doi.org/10.4211/hs.9f981ae4e68b4f529cdd7a5c9013e27e).



## 1 Introduction

Soil moisture data is essential for scientific inquiry in a variety of research areas. This data enables scientists to characterize hydrological patterns (Greve and Seneviratne, 2015), quantify the influence of soil moisture on terrestrial carbon dynamics (van der Molen et al., 2011), identify trends in global climate variability (Seneviratne et al., 2013), analyse the response of ecosystems to moisture decline (Zhou et al., 2014), or detect the impact of

moisture on models of land-atmosphere interactions (May et al., 2016). The integrity of current soil moisture data is fundamental for a comprehensive understanding of the global water cycle (Al-Yaari et al., 2019).

The main sources of soil moisture data are in situ soil moisture measurements through monitoring networks such as the International Soil Moisture Network (ISMN, Dorigo et al.,

2011a) and satellite soil moisture measurements such as those provided by European Space Agency-Climate Change Initiative (ESA-CCI, Dorigo et al., 2017; Liu et al., 2011). Both measurement techniques can quantify regional-to-continental global soil moisture patterns and dynamics (Gruber et al., 2020).

*In situ* soil moisture measurements assess soil moisture within specific study sites at

specific soil depths (e.g., 0-5 cm). These measurements are fine-grained as soil moisture sensors have a small and localized footprint, and despite national and international networks they are limited in much the world (Fig. 1). Collection of *in situ* soil moisture data across large areas is expensive and time consuming; in many cases, logistical challenges such as limited funding for data collection and accessibility of soil moisture monitoring sites make it

impossible.

On the other hand, satellite soil moisture measurements collected in the form of microwave radiometry using L-band (~ 1.4-1.427 GHz) and C-band (~4-8 GHz) are more effective for larger regional-to-global soil moisture measurements (Mohanty et al., 2017). As

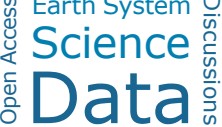

for most available *in situ* soil moisture measurements, satellite soil moisture datasets are

representative for the first 0-5 cm of soil depth. Unlike the fine-grained *in situ* measurements, satellite soil moisture datasets are available at the global scale in coarse-grained grids with spatial resolution ranging between 9km and 25km (Senanayake et al., 2019) and at the regional scale (e.g., the European continent) with a spatial resolution of 3km grids (Naz et al., 2020). A well-known satellite soil moisture dataset is collected by the European Space

Agency-Climate Change Initiative (ESA-CCI ).The ESA-CCI dataset contains more than 40 years of satellite soil moisture global grids (from the 1978 to 2019) with a spatial resolution of ~27km (Liu et al., 2011; Chung et al., 2018). This soil moisture dataset is a synthesis from multiple soil moisture sources and has been applied in long-term ecological and hydrological studies (Dorigo et al., 2017). The dataset covers a longer period of time compared with other

satellite-derived soil moisture datasets (e.g., Soil Moisture Active Passive [SMAP]) (Al-Yaari et al., 2019).

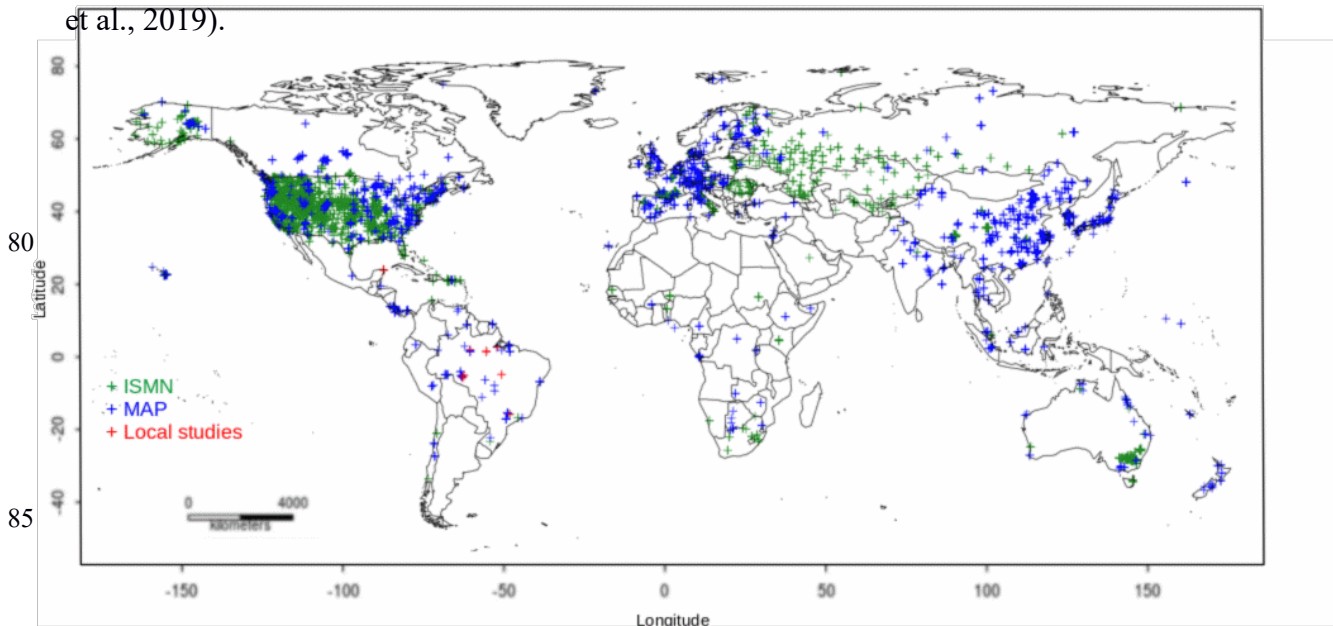



[Insert] Fig. 1 Spatial distribution of available *in situ* data (collected on the site) for validating soil moisture predictions. The ISMN (green), precipitation records (blue), soil moisture additional datasets from previous local studies (red).


Across large areas of the world, the ESA-CCI soil moisture data has been validated and calibrated against *in situ* soil moisture measurements (Al-Yaari et al., 2019; Dorigo et al., 2011a). In addition, there are continuing efforts to improve the spatial reliability of the satellite measurements (Gruber et al., 2017), resulting in new dataset versions (most recent v4.9). However, even the most recent version of ESA-CCI soil moisture data (i.e., v4.5-9) still suffers from a too coarse-grained spatial resolution and substantial spatial gaps in their spatial coverage (Llamas et al., 2020), making the data unsuitable to tackle problems such as quantifying the implications of soil moisture in water cycle across fine grained scales or across areas with spatial gaps. Scientists have developed empirical and physical modeling approaches for predicting missing satellite soil moisture data (Peng et al., 2017; Sabaghy et al., 2020) and for evaluating the errors in soil moisture satellite model predictions (Gruber et al., 2020). The spatial resolution and coverage of these recent studies is still an emergent challenge due to limited data across large areas of the world (e.g., extremely dry, extremely wet or frozen regions) as well as the signal excessive noise and saturation affecting the quality of satellite soil moisture records. Consequently, there is a need for developing alternative modeling approaches and their validation methods to fill the gaps of the ESA-CCI dataset, improving both the spatial resolution and the coverage.

In this paper we tackle this need by modeling and validating fine grained, gap free soil moisture predictions over the entire world. In doing so, we combine a pattern recognition technique called Kernel Weighted k-Nearest-Neighbors (or k-KNN, Hechenbichler and Schliep, 2004) with the use of independent covariate or prediction factors such as topographic parameters, bioclimatic features, and soil types. Our approach enables us to augment both



special resolution and coverage in the ESA-CCI dataset despite limited data in large areas of

the world.

k-KNN is a machine learning (ML) algorithm that has several benefits for predicting satellite soil moisture at the global scale.  First of all, k-KNN accounts for non-linearities (e.g., local and regional specific data patterns). Soil moisture data (as a dependent variable) can be predicted as a function of the spatial variability of environmental data (independent

variables) with different spatial resolution and coverage (Peng et al., 2017; Guevara and Vargas, 2019; Llamas et al., 2020). k-KNN can take advantage of the spatial autocorrelation of training data such as the relation between variance and distance between soil moisture observations (Llamas et al., 2020; Oliver and Webster, 2015) and use it as ancillary information when spatial coordinates (e.g., latitude and longitude) are considered in the

prediction approach (Hengl et al., 2018; Behrens et al., 2018; McBratney et al., 2003). Second, k-KNN can use kernel functions to weight the neighbors according to their distances. Finally, by including spatial coordinates in the predictions, k-KNN can consider geographical distances. In doing so, it is able to account for local and regional variability in the feature space: each predicted value is dependent on a unique combination of k neighbors in the

feature space that are weighted using kernel functions that can be different from one place to another (see Section 2.2 Refinement modeling).

We use a diverse set of independent covariates or prediction factors such as topographic parameters, bioclimatic features, and soil types to augment the prediction of soil moisture values with k-KNN. Topographic parameters are based on physical principles

related to the overall distribution of surface water across the landscape (Western et al., 2002; Moeslund et al., 2013; Mason et al., 2016). We generate the topographic parameters from digital terrain analysis. Digital terrain analysis involves calculations of land surface characteristics that depend on topography (e.g., terrain slope and aspect, Wilson, 2012). The impact of terrain parameters on spatial variability of satellite soil moisture is supported by

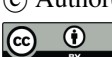



previous studies that have provided evidence of a topographic signal in satellite soil moisture measurements from local (Mason et al., 2016) to continental scales (Guevara and Vargas, 2019). Other studies derive terrain parameters from elevation data and use them to predict soil moisture across a gradient of hydrological conditions (Western et al., 2002). Topographic parameters have also been used for soil attribute predictions (Moore et al., 1993) and for soil

moisture mapping applications (Florinsky, 2016). All these studies suggest that topography (represented by multiple terrain parameters) is a useful predictor of surface soil moisture variability at the global scale. Different types of terrain parameters exist including elevation data structures, topographic wetness, overland flow, and potential incoming solar radiation among others. Elevation data structures (i.e., point elevation data, elevation contour lines, or

digital elevation models) quantitatively represent topographic variability and are the basis of digital terrain analysis (i.e., geomorphometry).  The topographic wetness index is a terrain parameter that characterizes areas where soil moisture increase by the effect of overland flow accumulation (Moore et al., 1993). Overland flow and potential incoming solar radiation are two important topographic drivers of the spatial distribution of soil moisture (Nicolai-Shaw et

al., 2015), its lags after precipitation events (McColl et al., 2017), and its role as a dominant control of plant productivity (Forkel et al., 2015). Bioclimatic features and soil types account for hydroclimatic and soil variability affecting soil moisture. We add bioclimatic features and soil type classes as additional prediction factors to our approach to determine if information beyond terrain parameters substantially improves soil moisture predictions. To validate our

dataset, we use independent field information from local studies (n=9, Vargas, 2012, Saleska et al., 2013), from the International Soil Moisture Network (ISMN, n=13376) and n=4909 in situ precipitation records (including n=171 sites across tropical areas poorly represented in the ISMN).

The contributions of this paper are twofold:  first, we integrate the k-KNN algorithm

and prediction factors into a modeling approach to predict fine grained, gap free soil moisture





data with a resolution of 15km, and second, we generate a new dataset that compliments the ESA-CCI dataset and is composed of soil moisture predictions from our modeling approach. With reference to our first contribution, we study the effectiveness of k-KNN to downscale satellite-derived soil moisture using two prediction factor datasets: a first dataset based only
170 on topographic parameters and a second based on topographic parameters, bioclimatic features, and soil types. We compare the accuracy of the two types of fine grained, gap free soil moisture models obtained using the two prediction factor datasets respectively. The comparison allows us to assess the impact of the individual prediction factors. Specifically, we address the impact of topographic parameters versus bioclimatic features and soil types.

175 Previous studies have used a variety of prediction factors for soil moisture, including vegetation indexes (from optical imagery), climate information (Alemohammad et al., 2018), chloropeth maps (i.e., land use and land-forms), thermal data and soil information to improve the spatial resolution and coverage of soil moisture gridded datasets (Naz et al., 2020, Peng et al., 2017). In contrast to past efforts, our solution uses a comprehensive set of factors for
180 predicting satellite soil moisture data and independently test the model with *in situ* soil moisture data. Our approach is computationally less expensive and prevents potential spurious correlations when predicted soil moisture estimates are compared with climate, vegetation, or soil information. With reference to our second contribution, we generate a dataset complementary to the ESA-CCI soil moisture dataset that uses the comprise gap free
185 global mean annual soil moisture predictions for 28 years (1991-2018) across a 15km grids (note that ESA-CCI has a grid of 27km). Our soil moisture dataset can be used for identifying spatial and temporal patterns of soil moisture and its contributions to climate and vegetation feedbacks. The soil moisture predictions, the field soil moisture validation dataset, and the set of prediction factors for soil moisture are available at (Guevara et al 2020).


## 2 Methodology





Our prediction approach has four key steps: First, we define two different datasets of prediction factors with a 15km global grid resolution: a dataset consisting only of terrain parameters and a different dataset combining terrain parameters, bioclimatic features, and soil type classes (Section 2.1). Second, we build prediction models by feeding the prediction factors and ESA-CCI satellite soil moisture data to the k-KNN algorithm and using cross validation for selecting the best models (Section 2.2). Third, we bootstrap the parameters to assess variances of soil moisture predictions (Sections 2.3). Last, we validate our best predictions against independent *in situ* soil moisture measurements when they are available (Section 2.4).


## 2.1 Datasets of Prediction Factors

We generate and test two different datasets of prediction factors with a 15km grid resolution: (a) a dataset of only digital terrain parameters and (b) a more complex dataset that uses digital terrain parameters, static bioclimatic features, and soil type information. The second dataset allows us to differentiate between the impact of terrain parameters in isolation versus the terrain parameters when augmented with static bioclimatic features and soil type information. The values of prediction factors are generated to overlap with the central coordinates (latitude and longitude) of the original ESA-CCI soil moisture pixels (Guevara and Vargas, 2019).

Digital terrain parameters (described in Fig. 2) are derived from a global digital elevation model using SAGA-GIS (System for Automated Geoscientific Analysis-GIS) (Conrad et al., 2015). The source of elevation data is a radar based digital elevation model (Becker et al., 2009). This digital elevation model is prepared by Hengl et al., (2017) and it is re-sampled (along with bioclimatic features and soil type classes) for this present study to a spatial resolution of 15km grids across the world. We consider these terrain parameters: (a) terrain aspect (aspect), (b) specific catchment area (carea), (c) channel network base level (chnl base), (d) distance to channel network (chnl dist), (e) flow convergence index (convergence), (f) horizontal curvature (hcurv), (g) digital elevation model (land), (h) length-





slope factor (lsfactor), (i) relative slope position (rsp), (j) analytical hillshade (shade), (k)
smoothed elevation (sinks), (l) terrain slope (slope), (m) valley depth index (vall depth), (n)

vertical curvature (vcurv), and (o) topographic wetness index (wetness). The parameters are
presented in Fig. 2. The detailed description and units of the parameters can be found in
Guevara and Vargas (2019).





[Insert] Fig. 2 Digital terrain parameters used as prediction factors for soil moisture. These parameters are derived from a digital elevation model using SAGA-GIS. These terrain parameters are standardized by centering their means in zero by a variance unit for improving visualization purposes. Legend: (a) terrain aspect (aspect), (b) specific catchment area (carea), (c) channel network base level (chnl base), (d) distance to channel network (chnl

dist), (e) flow convergence index (convergence), (f) horizontal curvature (hcurv), (g) digital elevation model (land), (h) length-slope factor (lsfactor), (i) relative slope position (rsp), (j) analytical hillshade (shade), (k) smoothed elevation (sinks), (l) terrain slope (slope), (m) valley depth index (vall depth), (n) vertical curvature (vcurv), and (o) topographic wetness index (wetness). See detailed description and units at (Guevara and Vargas, 2019).


Static bioclimatic features are extracted from the Food and Agriculture Organization Global Agro-Ecological Zones project (FAO, 2010, baseline period 1961-1990) to account for hydroclimatic variability. As soil type information, we include soil water retention

capacity classes (1 = 150 mm water per m of the soil unit, 2 = 125 mm, 3 = 100 mm, 4 = 75 mm, 5 = 50 mm, 6 = 15 mm, 7 = 0 mm) from the Re-gridded Harmonized World Soil Database v1.2 (Wieder et al., 2014) to account for soil type variability in our prediction framework.

For each pixel with available soil moisture values in the ESA-CCI dataset, we augment

the spatial coordinates (i.e., latitude and longitude) and soil moisture value by adding the tuple of the 15 terrain parameters for the first dataset, and the tuple of the 15 terrain parameters, the 19 bioclimatic features, and the soil type classes for the second dataset. The pixels without soil moisture values become our prediction targets. Because the prediction factor datasets have a 15km resolution while the ESA-CCI soil moisture pixels haver a 27km

resolution, we preprocess each prediction factor dataset to extract the values to the



corresponding locations of the ESA-CCI pixels. By overlapping the original ESA-CCI dataset with one of the two prediction factor datasets and extracting the prediction factor values for the ESA-CCI pixel centers, we generate two augmented ESA-CCI datasets. A similar method was initially used for the conterminous United States only (Guevara and Vargas, 2019). Here

we extend the method to the entire world.  In our mapping, we leverage observations from other work outlining the positive impact of spatial structure (e.g., spatial distances and autocorrelation) on soil attribute predictions (e.g., soil moisture) (see spatial coordinate maps in Appendix A) (Llamas et al., 2020; Møller et al. 2020; Hengl et al. 2018; Behrens et al., 2018; McBratney et al., 2003; Oliver and Webster, 2015).  We include spatial coordinates in

our modeling framework (described in Section 2.2) to account for the spatial structure of the ESA-CCI training data. To this end, we use spatial coordinates at multiple oblique angles as suggested by recent work (Møller et al., 2020, Appendix A). This preprocessing is done using open source R software functionalities for geographical information systems (R Core Team 2020, Hijmans, 2019).


## 2.2 Building Prediction Models

To build prediction models of the soil moisture at a finer spatial resolution (15km) than the

original ESA-CCI dataset  (27km), we use the kernel-based method for pattern recognition known as k-KNN (Hechenbichler and Schliep, 2004). We observe that the relationships among spatial coordinates, soil moisture values, terrain parameters, bioclimatic classes, and soil types are not linear. For example, south slope areas tend to be dryer than north slopes areas. Moreover, there is a contrasting feedback of soil moisture and precipitation between

humid and dry areas (e.g., between the Eastern and Western of the United States, Tuttle and



Salvucci, 2016). We use k-KNN because it allows us to account for the non-linear feedback while providing a simple and fast prediction solution.

The k-KNN algorithm has two main settings: (a) the parameter k that determines the number of neighbors from which information is considered for prediction, and (b) a kernel
function that converts distances among neighbors into weights, so the farther the neighbor, the smaller the weight it will be assigned. We consider k neighbors with k ranging from two to 50 soil moisture pixels and with close spatial coordinates and similar prediction factors. In the case of the first prediction factor dataset (i.e., only digital terrain parameters), distances among neighbors are computed among spatial coordinates and terrain parameters; in the case
of the second dataset (i.e., digital terrain parameters, static bioclimatic features, and soil type classes), distances among neighbors are computed among spatial coordinates, terrain parameters, static bioclimatic features, and soil type classes. The similarity among neighbors is measured with the Minkowski distance (i.e., the statistical average of the neighbors' values difference). We consider six different kernel functions (i.e., Rectangular, Triangular,
Epanechnikov, Gaussian, Rank, and Optimal).

Using the two augmented ESA-CCI datasets obtained by overlapping the original ESA-CCI dataset with one of the two prediction factor datasets and extracting the prediction factor values for the ESA-CCI pixel centers (from Section 2.1), we generate two sets of 28 prediction models, one for each of the 28 years (i.e., 1991-2018) in the ESA-CCI soil
moisture dataset (v4.5). We feed the augmented ESA-CCI datasets into the k-KNN algorithm and search for the most effective k neighbors' values and kernel functions. To this end, we use ten-cross validation to select the values of the k neighbors among the 48 possible values (i.e., k ranted from 2 to 50) and the kernel function from these six kernel functions (i.e., Rectangular, Triangular, Epanechnikov, Gaussian, Rank, and Optimal). We use cross-
validation as a re-sampling technique because it can prevent overfitting in ML methods such as k-KNN and can generate multiple sets of independent model residuals to evaluate the



stability of prediction outcomes. The use of cross-validation for searching for the most effective k neighbors' values and kernel function requires us to randomly create multiple independent training and testing datasets. Training and testing datasets generated from one of

our augmented ESA-CCI datasets are disjoined; training data is used for building the models, and testing data is used only for quantifying model residuals and evaluating soil moisture predictions.

      As our cross-validation indicators (i.e., information criteria about prediction), we use Pearson correlation coefficient (r) and the root mean squared error (RMSE) for each one of

the prediction models. For each year we select the model whose combination of k and kernel function has highest r and lowest RMSE. We use the model to predict annual mean global soil moisture across 15km global grids.

## 2.3 Assessing variances of model predictions

We study three sources of modeling variance. First, we assess the sensitivity of the prediction models to variations in available training data over the entire world. Second, we assess the relevance of the spatial coordinates and different prediction factors by rebuilding the models using the k-KNN algorithm with and without each prediction factor, once again over the entire world. Third, we assess the effectiveness of the k-KNN algorithm across selected areas

of the world with fewer data available for training the prediction models and with different environmental and climate gradients.

      To assess the sensitivity of the prediction models to variations in training data, we compute the variance of our soil moisture predictions as surrogates of model-based uncertainty. We rebuild the prediction models setting the k-KNN algorithm to use different

random subsets of available pixels (n=1,000) and 10-fold repeated cross-validation (n=10) to quantify the variance of soil moisture predictions. This model variance enables us to identify





geographical areas with high or low model uncertainty associated with the sensitivity of prediction models to random variations in training data.

To assess the relevance of the different prediction factors, we use the r and RMSE of modeling with all prediction factors as reference, and we compare the r and RMSE with the r and RMSE values of modeling without each one of the prediction factors. We test the sensitivity of the spatial coordinates and each prediction factor (i.e., terrain parameters, bioclimatic features, and soil type classes) by systematically leaving out one prediction factor at a time and repeating our k-KNN algorithm and its respective cross-validation. This process
is repeated ten times for each prediction factor to capture a variance estimate. This empirical validation approach provides empirical insights of the relative importance of prediction factors for the k-KNN algorithm predicting soil moisture at the global scale.

To assess the effectiveness of the k-KNN algorithm across specific areas of the world, we first test the k-KNN algorithm under tropical areas (Appendix B) with low availability of
data to train prediction models (e.g., higher distances between k neighbors) and homogeneous environmental and climate conditions (e.g., higher water content aboveground than below ground).  We extract the limits of tropical areas from the Global Agro-Ecological Zones project (FAO, 2010, baseline period 1961-1990, described in section 2.1). Second, we test the k-KNN algorithm using only the available ESA-CCI data across counties with large
heterogenous environmental and climate gradients such as Canada, Australia, and Mexico. We generate new training, testing, and prediction factors datasets for these countries using geopolitical limits provided by the global administrative maps initiative (GADM, 2018). We use the resulting model predictions to explore modeling consistency in terms of r and RMSE values across the selected areas and to visualize spatial patterns between the ESA-CCI soil
moisture dataset and our soil moisture predictions.

**2.4 Validation against independent *in situ* data**



When *in situ* soil moisture data (i.e., ISMN) and *in situ* precipitation records (as an alternative validation approach across areas with low availability of *in situ* soil moisture data) are

available in the ISMN dataset (Dorigo et al., 2011a, 2017), we validate the ESA-CCI dataset and our predictions against those local soil moisture data reported in ISMN for each year. Additionally, we compare soil moisture trends (i.e., changes in soil moisture over time) across time by comparing either *in situ* soil moisture or the ESA-CCI with our predictions.

To validate the ESA-CCI dataset and our predictions against the available *in situ* soil

moisture data, we augment the original ISMN datasets with its *in situ* soil moisture data for the years 1991-2018 (see Section 5) built from its 8,080 tables. In doing so, because tropical areas are poorly represented in the ISMN, we first further extend ISMN dataset by including 10 more stations with *in situ* soil moisture data from literature reviews that are distributed in open access data repositories: one site in a tropical forest of Mexico with data from 2006-

2008 (Vargas, 2012) and nine sites across Brazil's tropical forests with data from 1999-2006 (Saleska, et al., 2013). Second, across areas of the world with low availability of *in situ* soil moisture information, as it is the case for tropical areas (Fig. 1), we use *in situ* records of annual precipitations (n=4909) including 171 sites (years 2008 to 2018) from the global soil respiration database (Bond-Lamberty and Thomson, 2018), and we compute the correlation

between satellite soil moisture data and *in situ* precipitation records when no other independent *in situ* soil moisture data is available. We add these correlations to the augmented ISMN dataset. The use of precipitation data for areas of the world where no *in situ* soil moisture validation data is supported by work of Gruber et al., (2020).

Subsequently, we extract the values of the ESA-CCI soil moisture dataset and the

values of our soil moisture predictions for each one of the locations reported in our augmented ISMN dataset of *in situ* soil moisture values. Any potential bias associated with the data in our augmented ISMN dataset (e.g., stations with low number of records) has potentially the same impact on the validation results of the three datasets (ESA-CCI and our





two prediction datasets). In other words, we assume biases are randomly distributed across all

observations, and thus they are not accounted for the outcome of our comparisons. We

summarize the validation results in a target diagram to illustrate the accuracy of our soil

moisture predictions. The target diagram (presented in Appendix C, Jolliff et al., 2009;

Gruber et al., 2020) shows the relation between the variance and magnitude of errors (e.g.,

unbiased root mean squared error or ubRMSE) (a) between the ESA-CCI and the augmented

ISMN dataset and (b) between our predictions and the augmented ISMN dataset.

To compare trends in soil moisture over time for areas for which we have *in situ* data,

we perform a non-parametric (median-based) trend detection test (i.e., Theil-Sen estimator) to

compare soil moisture trends at the locations of the augmented ISMN dataset. This trend

detection is done by calculating the median value of the slopes and intercepts of all possible

combinations of pairs of points in the relationship of soil moisture (response) and time

(explanatory variable). This resulting median slope and intercept estimates are unbiased and

resistant to outliers (Kunsch, 1989).

For those areas in which the ISMN dataset has multiple gaps, we rely on the ESA-CCI

and our prediction datasets to generate a map of soil moisture trends. To this end, we apply a

pixel-wise trend detection test to the ESA-CCI and prediction datasets to search for possible

breakpoints (i.e., significant changes in soil moisture over time). We consider two regression

parameters (i.e., slopes and intercepts) before and after any possible breakpoint to detect

trends; in all the tests, a minimum of four years is required between breakpoints for detecting

trends. To provide our study with robust trend detection estimates, we do not consider

segments between breakpoints with less than eight observations. (Forkel et al., 2013, 2015).

## 3 Results

In our assessment of the results, we first discuss the statistical description of the observed and

modeled soil moisture datasets (Section 3.1). Second, we present the sensitivity of the



prediction models and the way they are generated to variations in available datasets (Section

3.2). Third, we measure the relevance of the different prediction factors by rebuilding the

models using the k-KNN algorithm with and without one prediction factor at the time over

the entire world (Section 3.3). Finally, we summarize results on soil moisture for models that

are trained on regions for which augmented ISMN datasets exist (Section 3.4) and results on

soil moisture for models that are trained on regions for which augmented ISMN datasets do

not exist and thus we use either ESA-CCI or our predictions as alternative datasets (Section

3.5).

### 3.1 Descriptive Statistics

We first assess the statistical distributions of the observed ESA-CCI dataset, our soil moisture

model predictions using the k-KNN algorithm, and the augmented ISMN dataset (Fig. 3).

Comparing the statistical distribution between observed datasets (i.e., ESE-CCI and ISMN

datasets) and our modeled soil moisture datasets allows us to identify if modeled soil

moisture falls within the expected range of observed soil moisture values. The statistical

distribution among different soil moisture datasets can be compared in terms of differences in

the mean and standard deviation. We present the mean and standard deviation of the ESA-

CCI dataset, our modeled soil moisture predictions, and the augmented ISMN dataset only at

locations (latitude and longitude) where all datasets have an observation or a prediction. We

also restrain the period of time for our comparisons between 1991-2016, which is the period

of time with higher consistency of data availability for both the ESA-CCI dataset and the

augmented ISMN dataset.

When comparing the statistical distribution of the soil moisture datasets, we observe

that the ESA-CCI dataset has mean soil moisture values of 0.29 $m^3/m^3$ and a standard

deviation of 0.09 $m^3/m^3$. The modeled soil moisture predictions based only on digital terrain

parameters has mean soil moisture values of 0.24 $m^3/m^3$ and a standard deviation of 0.05





$m^3/m^3$. Modeled soil moisture predictions based on digital terrain parameters, bioclimatic features, and soil type classes show mean soil moisture value is 0.24 $m^3/m^3$ and a standard deviation is 0.05 $m^3/m^3$. The augmented ISMN dataset shows a larger range of soil moisture values (Fig. 3) comparing all datasets: the dataset values show a mean of 0.25 $m^3/m^3$ and a standard deviation 0.07 $m^3/m^3$. We have two key observations. First, we observe a consistent

statistical distribution comparing the statistical distribution of the augmented ISMN compared with the statistical distribution of the ESA-CCI dataset (Fig. 3). Second, and more importantly, the mean and standard deviation of our modeled soil moisture predictions based on terrain parameters only and based on terrain parameters, bioclimatic features, and soil type





classes as prediction factors show similar agreement with the means and standard deviations

of both ESA-CCI and augmented ISMN datasets.

[Insert] Fig. 3 Statistical distribution of the ESA-CCI soil moisture dataset (red), the

predictions of soil moisture using the k-KNN algorithm (gray and green) and the augmented

ISMN dataset (black). The lines represent the values of each dataset at the locations of all

datasets exist (locations reported in the augmented ISMN).


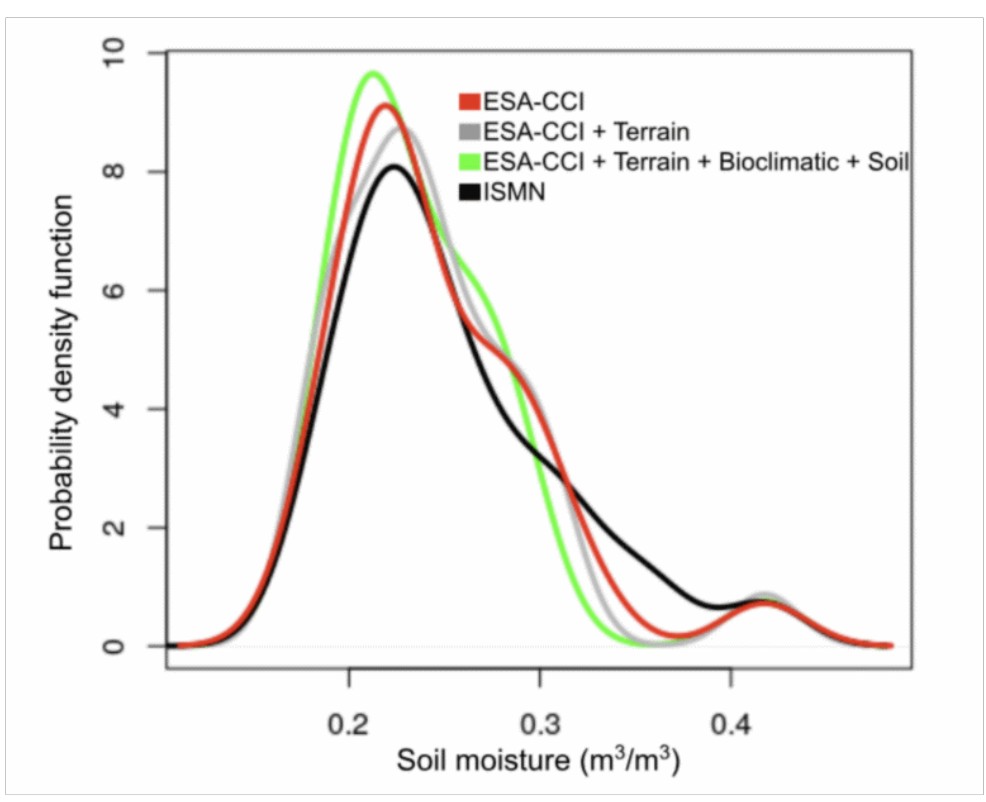



## 3.2 Prediction Sensitivity for Different Datasets

We evaluate r and RMSE for 12,040 cross-validated soil moisture models. The number of models is defined as follows. For each year (n=28) we build a model with all prediction factors (n=42) and assess the variance of 10 model replicas based on different random data subsets (n – 10% of data). We repeat the same process for each year leaving out each one of the prediction factors at the time and assess the prediction sensitivity for different datasets as explained in Section 2.3. We compute the r and RMSE between observation and model prediction datasets. Our observations are soil moisture values from the ESA-CCI dataset and from the augmented ISMN as generated in Section 2.4. Our prediction factors datasets (defined in Section 2.1) are the basis to generate: (a) the soil moisture predictions based on terrain parameters only and (b) the soil moisture predictions based on terrain parameters, bioclimatic features, and soil type classes.

We first report results for the entire world using ESA-CCI as training dataset for building prediction models and repeated cross validation for assessing the accuracy of the model predictions (described in Section 2.2). The cross-validated r of soil moisture predictions based on digital terrain parameters only ranges from 0.69 to 0.81 across years (1978-2019). The RMSE ranges from 0.03 to 0.04 $m^3/m^3$. The soil moisture predictions based on terrain parameters, bioclimatic features, and soil type classes have slightly higher correlation between observed and predicted soil moisture values (ranging between 0.78 and 0.85) and slightly lower RMSE values (ranging from 0.02 to 0.04 $m^3/m^3$). Note that each soil moisture prediction contains a cross validation accuracy report (see Section 5). The small variations of r and RMSE indicate a reliable prediction capacity of our models.

For the entire world once again, we assess the sensitivity of our predictions (described in Section 2.3) in terms of the models' prediction variance, which ranges from <0.001 to 0.18 $m^3/m^3$. This prediction variance is higher in areas with lower availability of training data




from the ESA-CCI (e.g., across the tropical areas and coastal areas). These variances also serve as surrogates for uncertainty; each file containing a soil moisture prediction model includes a file with a soil moisture prediction variance (see Section 5 data availability). For example, for the year 2018 (Fig. 4), soil moisture predictions varied between ~0.001 and

~0.45 $m^3/m^3$ while the prediction variances range from ~0.001 to 0.14 $m^3/m^3$, indicating a broader variability around the predicted values. Larger prediction variances are the combined result of both the higher possible values of soil moisture and the limited sample size within the ESA-CCI to train the prediction models, such as in tropical areas dominated with dense vegetation.

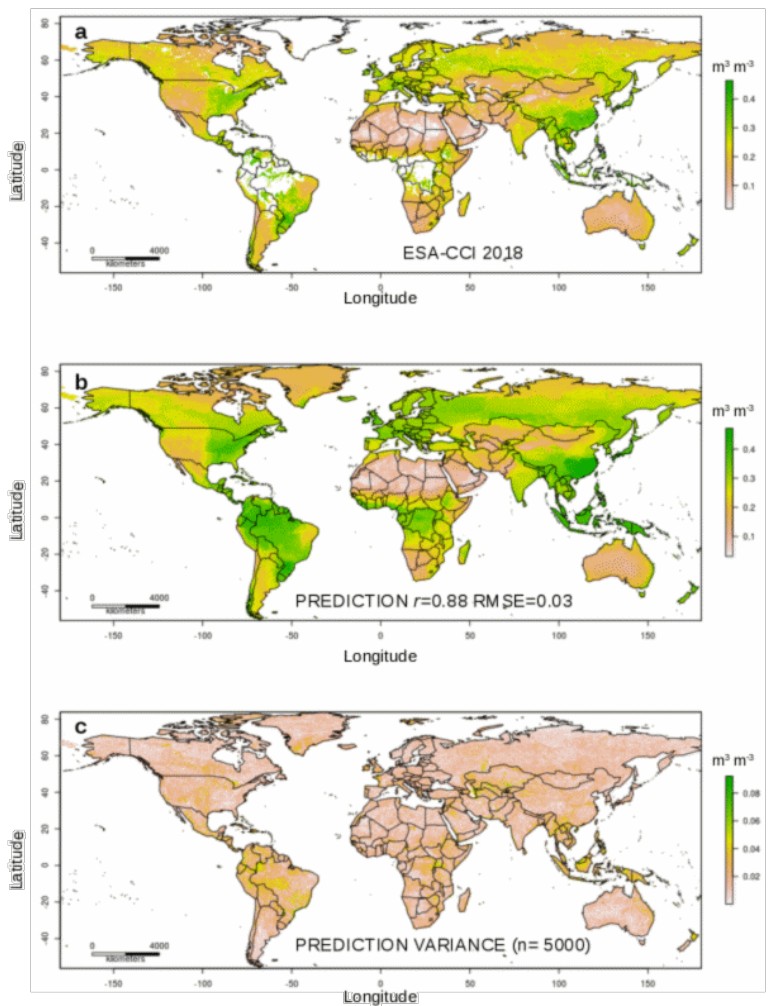


[Insert] Fig. 4 Soil moisture prediction and prediction variance (5000 x 5), snapshot 2018, showing the mean values of the ESA-CCI for 2018 (a), the resulting k-KNN prediction (b), and the prediction variance (c).


We provide an example of the sensitivity of our models across tropical areas with low available data for training the models as described in Section 2.3. For tropical areas of the world

with limited information in the ESA-CCI datasets, the cross-validated results of the model
predictions showed r values around 0.62 and RMSE values around 0.03 m³/m³ using terrain
parameters and soil type classes (Appendix B). We find that the model predictions based only
in the limited ESA-CCI soil moisture information available across tropical areas (Appendix B)
shows a similar prediction variance compared with the model predictions for the entire world,
with values from <0.001 to <0.12 m³/m³ (Appendix B). These result support the effectiveness
of our approach across areas with lower availability of information to train the k-KNN
algorithm.

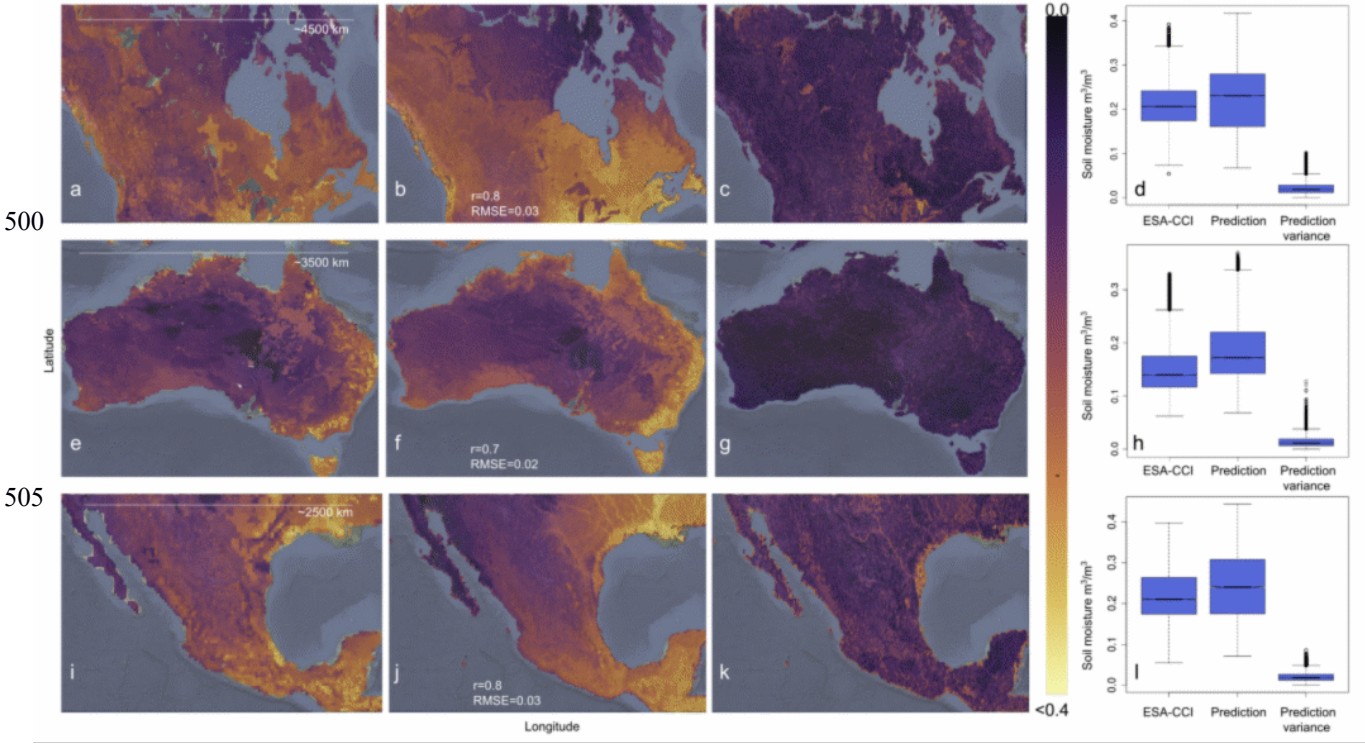


[Insert] Fig. 5 Examples of downscaled annual mean soil moisture across specific countries.
Prediction of soil moisture, prediction variance and training data from the ESA-CCI across
Canada (CAN; a-c), and their respective boxplots (showing their statistical distribution) for





the year 2018 (d). Prediction of soil moisture, prediction variance and training data from the

ESA-CCI across Australia (AUS; e-g), and their respective boxplots for the year 2018 (h). Prediction of soil moisture, prediction variance and training data from the ESA-CCI across Mexico (MEX; i-k), and their respective boxplots for the year 2018 (l).

We additionally assess the sensitivity of the model predictions across areas of the world with heterogeneous environmental and climate gradients (i.e., geographical extent of countries such as Mexico, Canada and Australia), generated as described in Section 2.3. The ESA-CCI has a relatively better spatial coverage across these countries (Fig. 5) compared with tropical areas (Appendix B) but still with a lower amount of training data compared with

models generated for the entire world. Comparing our soil moisture predictions across 15km grids with the original ESA-CCI soil moisture dataset at 27km grids (Fig. 8) for these areas, we observe that our soil moisture predictions have consistently higher maximum values (>0.04 $m^3m^{-3}$) than the original ESA-CCI soil moisture dataset (<0.4 $m^3m^{-3}$) (Fig. 5). We observe consistent modeling accuracy across these countries and across the entire world (in

all cases r values >0.6 and RMSE values around 0.04 $m^3m^{-3}$).

The last two sets of results for tropical areas with low available data and areas of the world with heterogeneous environmental and climate gradients support the effectiveness of our approach across areas exhibiting unfeasible data collection and heterogenous data characteristics respectively. The flexibility of our prediction models to generate consistent

results on a country-specific basis could be supported by the use of country specific information (e.g., topographic, bioclimatic, and soil information) to predict soil moisture with higher spatial resolution (<15km grids) in future research.

### 3.3 Relevance of the different prediction factors



Across the entire world, we assess the relevance of the different prediction factors defined in
Section 2.1 (i.e., prediction factors from terrain parameters, bioclimatic features, and soil type
classes) by rebuilding the prediction models using the k-KNN algorithm and removing one
prediction factor at a time. By systematically removing one prediction factor at a time and
using repeated 10-fold cross-validation (n=10), we can measure the prediction factor impact

on the accuracy of each model generated for each year using the k-KNN algorithm (Fig. 6).
To this end, we compare the cross-validation results (r and RMSE values) of each new model
against a reference model that we build by using all prediction factors. Each soil moisture
prediction using all prediction factors for each year is accompanied by a reference accuracy
report containing the cross-validation results (see Section 5 Data availability). We sort the

relevance of prediction factors based on the impact of their absence on the cross-validation
results (r and RMSE values), compared with the reference models (using all prediction
factors) across each year. Specifically, for each year (1991-2018) and for each factor that is
removed at the time (42 factors), we repeat ten times the cross validation as explained in
Section 2.2 and compute the mean accuracy. For each factor, we count the number of times

when the absence of that factor causes a higher r and a lower RMSE compared with the mean
accuracy of the reference model generated for each of the 28 years. Across the years we
count the number of positive and negative impacts and show the proportion of times (or
impact rate) when the absence of each prediction factor results in a higher accuracy (i.e.,
higher r and lower RMSE) versus the proportion of times when the absence results in a lower

accuracy (i.e., lower  r and higher RMSE)  (Fig. 6).



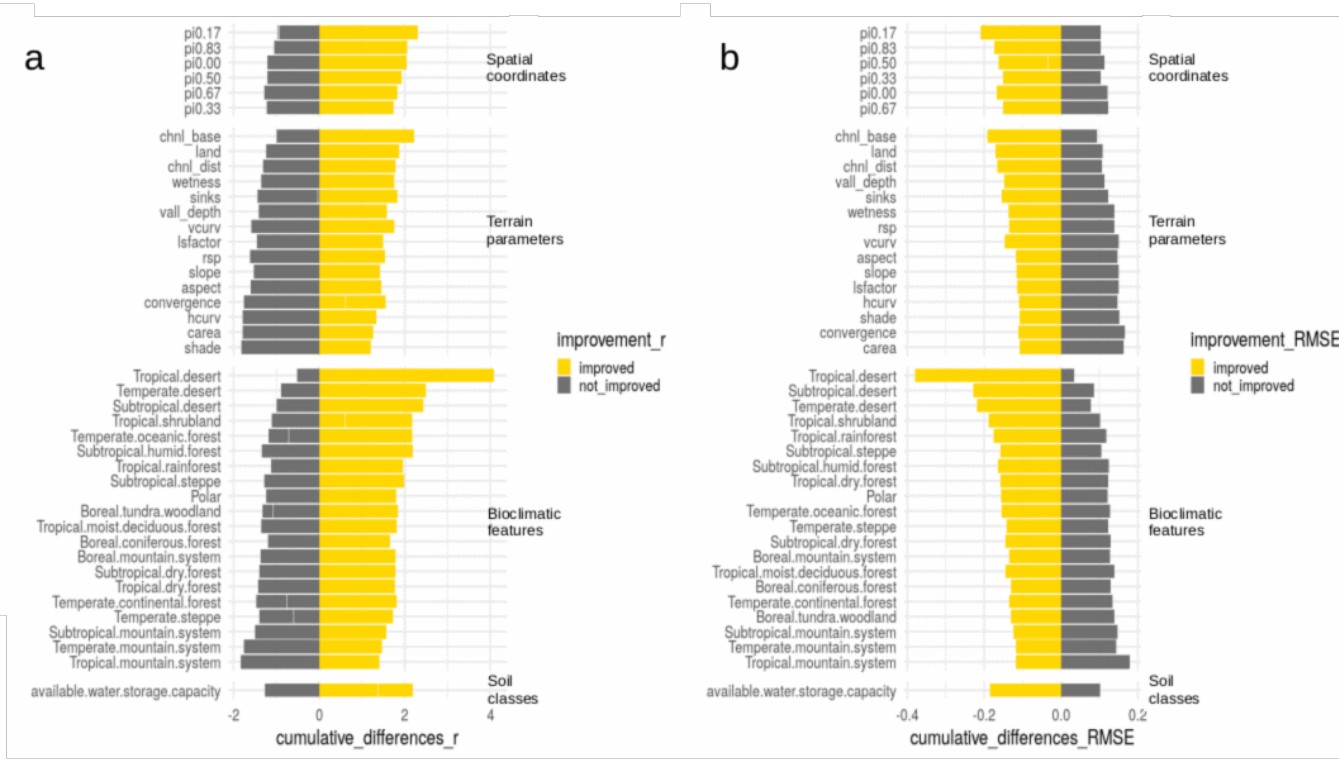

[Insert] Fig. 6  Impact of each factor on (a) r and (b) RMSE values across years.  The factors with code: pi0.00, pi0.17, pi0.33, pi0.50, pi0.67 and pi0.83 are the spatial coordinates rotated at multiple angles shown in Appendix A.The rest of the factors are the digital terrain parameters used to predict the ESA-CCI annual means as they are shown in Figure 3 and described by Guevara and Vargas (2019): aspect: terrain aspect, carea: specific catchment area, chnl base: channel network base level, chnl dist: distance to channel network, convergence: flow convergence index, hcurv: horizontal curvature, land: digital elevation






model, lsfactor: length-slope factor, rsp: relative slope position, shade: analytical hillshade,

sinks: smoothed elevation, slope: terrain slope, vall depth: valley depth index, vcurv: vertical

curvature, wetness: topographic wetness index. The bioclimatic features in: a) tropical, b)

subtropical, c) temperate or e) boreal environments are represented by binomial variables (0-

1). These variables are extracted by the Food and Agriculture Organization Global Agro-

Ecological Zones project. The available water storage capacity variable is represented by

continuous classes available thanks to the Re-gridded Harmonized World Soil Database.

We sort the relevance of prediction factors based on the impact of their absence on the

cross validation results (r and RMSE values), compared with the reference models (using all

prediction factors) across each year: for r (Fig. 6a) a negative impact rate of a factor means

that the model tends to improve in accuracy (in terms of higher r) when including that factor

and vice versa a positive impact means that the correlation increased when the factor is

removed. In contrast, for RMSE (Fig. 6b) a positive impact means that the model will tend to

improve accuracy (in terms of lower RMSE) when including that factor, and vice versa (a

negative impact means that the error decreased when the factor is removed).

We observe that spatial coordinates in rotated angles ranging between 17% and 83%

degrees (Appendix A) are coordinates with positive impact on r and RMSE results across

years (Fig. 6). Considered each year in isolation, we observe that values for r and RMSE are

consistent across individual years. In Figure 7 we present the values for r and RMSE for 2018

as a representative case. In 2018, we observe that spatial coordinates rotated in an oblique

angle between 33% to 50% degrees (variables pi0.33 and pi0.50, Appendix A) have high

impact on r or RMSE values (Fig. 7a).

Across all years, we find bioclimatic features have a higher impact on r or RMSE

values, followed by terrain parameters and soil classes (Fig. 6), which supports further





findings in our validation against *in situ* soil moisture data contained in the augmented ISMN
(in Section 3.4). We find that the use of spatial coordinates has similar impact on r and
RMSE values compared with terrain parameters or soil type classes (Fig. 6), We observe
slightly higher (but statistically similar) impact of bioclimatic features in cross validation

results compared with terrain parameters (Fig. 6). Bioclimatic features indicating presence or
absence (0/1 bionomial variable) of tropical, subtropical, or temperate desert (biological and
climatological) conditions are variables with high impact in the cross validation of prediction
models. The height between the base of  drainage networks channels to the closest highest
point in the ground (before elevation decreases again) (code in Fig. 6: chnl_base) or the

distance of each pixel to the closest drainage network channel (code in Fig. 6: chnl_dist) are
elevation (code in Fig. 6: land) derived terrain parameters with high impact on r and RMSE
across all years. We observe for our example with the year 2018 that terrain parameters such
as chnl_base and chnl_dist have higher impact on r and RMSE values consistently with our
analysis across all years (1991-2018). Bioclimatic features indicating the presence or absence

(0/1 bionomial variable) of temperate steppe climate conditions or the presence or absence of
tropical shrubland climate conditions become top prediction factors for soil moisture in this
specific year (2018, Fig. 7). The impact of terrain parameters have different impact for
predicting soil moisture variability depending of the average amount of water reaching the
soil (via precipitation and runoff or overlandflow) for each year, which is a process highly

dependent on bioclimatic conditions. Thus, we can expect to observe variations in the impact
of parameters to predict soil moisture across specific years (e.g., in extremely dry versus
extremely wet years).  We provide a variable importance plot for each year associated with
each soil moisture prediction (Section 5 Data availability).


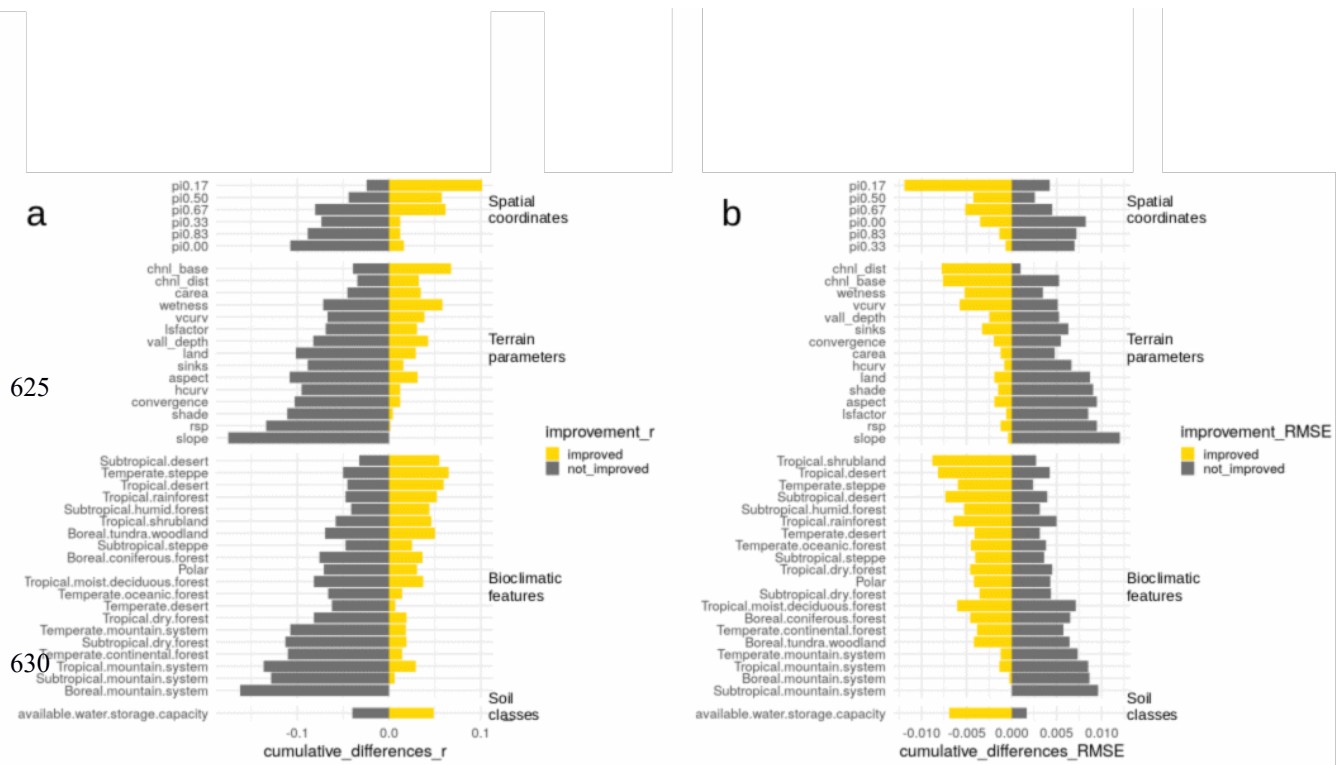

[Insert] Fig. 7 Impact of each factor on the (a) r and (b) RMSE values for the year 2018. The
factors named pi0.00, pi0.17, pi0.33, pi0.50, pi0.67 and pi0.83 are the spatial coordinates at
multiple angles shown in Appendix A.The digital terrain parameters are shown in Figure 3
and described by Guevara and Vargas (2019): aspect: terrain aspect, carea: specific
catchment area, chnl base: channel network base level, chnl dist: distance to channel network,
convergence: flow convergence index, hcurv: horizontal curvature, land: digital elevation
model, lsfactor: length-slope factor, rsp: relative slope position, shade: analytical hillshade,
sinks: smoothed elevation, slope: terrain slope, vall depth: valley depth index, vcurv: vertical
curvature, wetness: topographic wetness index. The bioclimatic features are divided in: a)
tropical, b) subtropical, c) temperate or e) boreal environments are represented by binomial
variables (0-1). These variables are extracted by the Food and Agriculture Organization
Global Agro-Ecological Zones project. The available water storage capacity variable is
represented by continuous classes available thanks to the Re-gridded Harmonized World Soil
Database.



[Insert] Fig. 8 Model evaluation plots (points vs grids). ISMN against the ESA-CCI (a),
ISMN against the predictions based on terrain analysis (b), and ISMN against the predictions
based on the model using bioclimatic and soil type classes (c). The panels below show the
correlation between soil moisture grids and in situ mean annual precipitation records and: in
situ precipitation against the ESA-CCI (d), in situ precipitation against predictions based on
terrain analysis (e) and in situ precipitation against predictions based on the model using
bioclimatic and soil type classes (f).



### 3.4 Soil moisture trained for region for which augmented ISMN datasets exist

To compare soil moisture values between our predictions and the augmented ISMN, we
followed two main steps. First, we assess the r and RMSE values between the ESA-CCI
dataset and our soil moisture predictions against in situ soil moisture using the augmented
ISMN. Second, we report changes of soil moisture over time using the augmented ISMN, the
ESA-CCI and our soil moisture predictions.

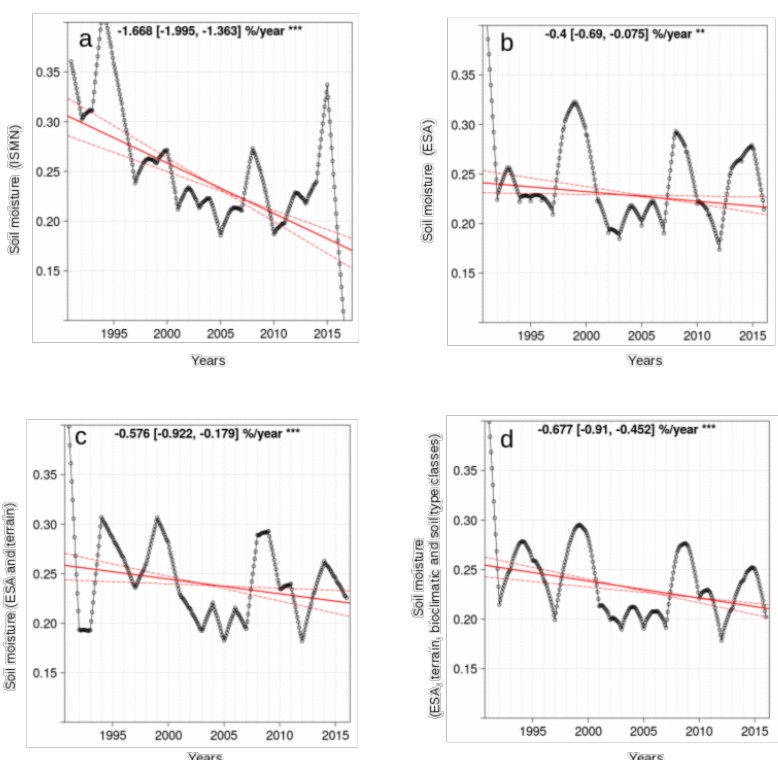

[Insert] Fig. 9 Negative soil moisture trends at the places of field stations in the ISMN. Trend
of ISMN (a), trend of the model based on the ESA-CCI dataset (b). Trend of the models
using terrain parameters (c) and trend of the models using bioclimatic and soil type classes
(e).





Comparing the correlation between *in situ* and gridded soil moisture datasets, we

observe that the correlation of the ESA-CCI (v4.5) with the augmented ISMN across the

world is lower compared to the correlation between soil moisture predictions based on digital

terrain analysis with the ISMN or soil moisture predictions adding bioclimatic and soil type

classes (Fig. 8).  The r values show a mean of 0.50 between the ISMN and the ESA-CCI, the

predictions based on digital terrain parameters show an r value of 0.59, and the predictions

including bioclimatic and soil type classes show an r value of 0.65. Similar levels of RMSE

against the ISMN are found with the models using bioclimatic and soil type classes (~0.05

$m^3/m^3$ ) or models using only terrain parameters  (~0.05 $m^3/m^3$ ). When comparing the ISMN

and the ESA-CCI, we observe a mean RMSE of 0.09 $m^3/m^3$ (Fig. 8, a-c). The target diagram

presented in Appendix C is useful to visualize the improvement of our approach against the

original ESA-CCI soil moisture dataset.

Across all analyzed years, our global soil moisture predictions represent an

improvement as they reduce bias when compared with the ISMN data and in situ

precipitation records. The variance around the prediction error (e.g., the unbiased RMSE)

estimated against the augmented ISMN was also lower in our predictions compared with the

ESA-CCI soil moisture dataset (Appendix C).

We confirm the effectiveness of the k-KNN algorithm for modeling and predicting

soil moisture considering changes in soil moisture levels over time (soil moisture trends, Fig.

9). There is a consistent soil moisture decline over time across all soil moisture datasets (i.e.,

the augmented ISMN and the ESA-CCI datasets, the soil moisture predictions based on

digital terrain analysis, and the predictions using digital terrain analysis, bioclimatic and soil

type classes) at the specific locations of the augmented ISMN dataset (Fig. 1). Temporal

trends show a decline of global annual soil moisture levels using: a) data from the ISMN (-

1.7 [-1.9, -1.4]%, b) associated locations from the original ESA-CCI dataset (-0.4[-0.69, -





0.01]%), c) associated locations from predictions based on terrain parameters (-0.58[-0.92, -

0.17]%), and d) associated locations from predictions including bioclimatic and soil type

classes (-0.68[-0.91, -0.45]%). Supporting the effectiveness of the model predictions, all

datasets (observed and modeled soil moisture) show negative soil moisture trends at locations

where all datasets exist.

## 3.5 Soil moisture trained for region for which augmented ISMN datasets do not exist

To compare soil moisture values across the entire world we followed two main steps. First,

we assess soil moisture trends across areas with no available data in the augmented ISMN

using *in situ* precipitation data (Fig. 1 blue). Second, we assess changes over time across the

areas with available data in the ESA-CCI dataset. Third we assess changes of soil moisture

across the world using our soil moisture predictions.

Comparing the correlation of the ESA-CCI and our soil moisture predictions we

observe that our predictions are better correlated with *in situ* precipitation records across

areas with no available data in the augmented ISMN (Fig. 8, d-f). These results are consistent

restraining the comparison to tropical areas (r=0.31 to r=0.38), boreal areas (r=34 to r=41) or

temperate areas (r=40 to r=51) of the world. These result support the effectiveness of the

model predictions across areas with low available *in situ* soil moisture validation data.

By analyzing changes of soil moisture over time using the ESA-CCI dataset across

the entire world (when available), we observe significant soil moisture increase (positive

trend) over time across ~70 000km$^2$ of the global land area (>500 million km$^2$) using a

probability threshold of 0.05, with available data during 1991 and 2018. We also observe a

significant decline (negative trend) of soil moisture across 43740 km$^2$ of global land area

(Fig. 10, a-b). In contrast, across the entire world, soil moisture based on terrain parameters

shows >60 million km$^2$ of global land area with negative trends and 274147 km$^2$ with positive

trends (Fig. 8, c-d).

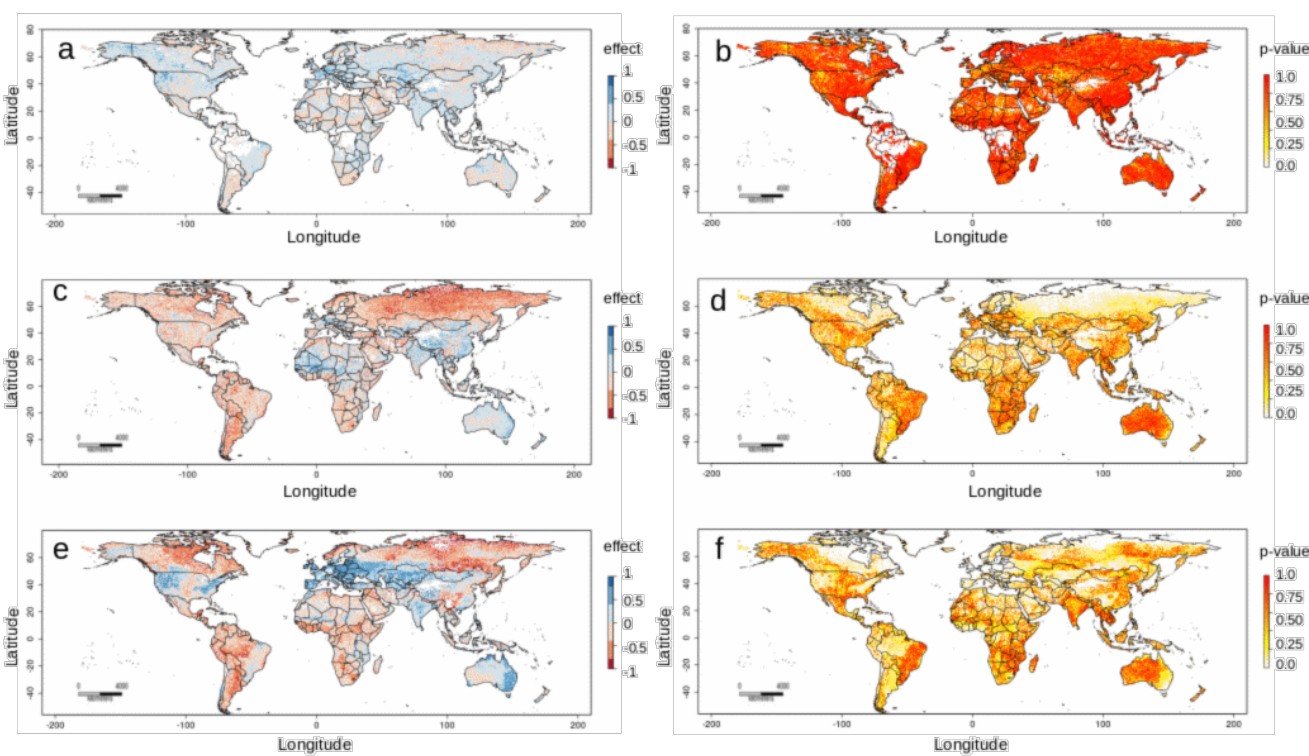


[Insert] Fig. 10 Trends of the ESA-CCI annual means (a) and their respective probability
values (b). Trends of the soil moisture predictions based on digital terrain parameters (c) and
their respective probability values (d). Trends of soil moisture predictions using terrain
parameters, bioclimatic and soil type features (e) and their respective probability values (f).




The soil moisture predictions based on terrain parameters, bioclimatic and soil type features showed significant negative trends (probability threshold <0.05) across 216 246 km$^2$ and positive trends across 85991 km$^2$ (Fig. 10, e-f) of global land area. Discrepancies between the ESA-CCI and our downscaled datasets are in part because our results predict soil moisture decline across areas with large gaps in the ESA-CCI, such as tropical areas. For example, with our soil moisture predictions we observe emergent negative trends of soil moisture across tropical rain forests of the Amazon basin and the Congo region.

## 4 Discussion

We present a regression approach coupling k-KNN and digital terrain analysis for improving the spatial resolution of ESA-CCI satellite soil moisture estimates by nearly 50% and providing a gap-free global annual mean soil moisture dataset (with associated uncertainty) for years 1991-2018. In this section we interpret and describe the significance of the new soil moisture datasets (based on terrain continuous parameters, soil and climate classes) in light of what was already known thanks to state-of-the-art satellite soil moisture (e.g., from the ESA-CCI) about the research problem of accuracy, coarse granularity and spatial gaps of soil moisture information at the global scale (incomplete global coverage).

We outline the key findings and insights organized in terms of their impact. First, we highlight the main improvements of the new soil moisture dataset against the ESA-CCI soil moisture product. Second, we discuss the role of terrain parameters in the accuracy of the new generated dataset. Third, we discuss emergent soil moisture trends before and after taking our new datasets into consideration. Fourth, we discuss potential sources of variance and discrepancy between soil moisture datasets (e.g., augmented ISMN, ESA-CCI, our predictions). Fifth, we provide information about the main limitations of the new dataset and sixth, we discuss opportunities for future work.



We highlight the main improvements of the new soil moisture dataset against the ESA-CCI soil moisture product. Our predictions of soil moisture against the ESA-CCI soil moisture product show an improvement in the reduction of bias when compared with *in situ* soil moisture datasets (i.e., with the ISMN, Fig. 8 and Appendix C). Improving the spatial resolution of satellite-derived soil moisture is an ongoing challenge that requires different approaches. For example, recent soil moisture remote sensing datasets (Entekhabi et al., 2010, Piles et al., 2019) are able to provide information across areas with spatial gaps in the ESA-CCI; however, only recent years have full soil moisture coverage (e.g., 2010 to date). Our results represent a long-term (1991-2018) and gap-free soil moisture dataset and represent a response to the need of alternative global-to-regional soil moisture datasets (An et al., 2016; Colliander et al., 2017b; Dorigo et al., 2011b; Minet et al., 2012; Mohanty et al., 2017; Yee et al., 2016). This dataset has implications for further analyses on soil moisture patterns (Berg and Sheffield, 2018), global hydrological models (Zhuo et al., 2016), climate change predictions (Samaniego et al., 2018), carbon cycling models (Green et al., 2019), and food security assessments (Mishra et al., 2019).

We now discuss the role of terrain parameters in the prediction accuracy of the new generated dataset. We demonstrated the role of topographic terrain parameters as a parsimonious and effective approach for downscaling satellite-derived soil moisture in terms of r (Fig. 6) or RMSE (Fig. 7). Terrain parameters are available nowadays with unprecedented levels of spatial resolution (e.g., meters) and our approach is potentially applicable to specific areas or countries (Fig. 5) and higher spatial resolution (Guevara and Vargas, 2019). These results (e.g., Fig. 5, Appendix B) support the value of terrain parameters as the basis for downscaling soil moisture satellite estimates in future research across specific areas or periods of time. The exclusive use of terrain parameters in our algorithm implementation (Section 2.2) can  help to reduce model complexity and computational expenses of more complex models using a extensive set of prediction factors



for representing soil variability (e.g., Hengl et al., 2017). A soil moisture dataset independent of bioclimatic and soil information is useful to prevent potential spurious correlations in

further studies. This is specifically important for studies dealing with the problem of interpreting machine learning frameworks or better understanding the use of data by the algorithms to generate accurate model predictions (Padarian et al., 2020, Ribeiro et al., 2016). In the other hand, predicting soil moisture considering tacit knowledge (i.e., expert opinion) on variable selection (e.g., combining manually multiple combinations of prediction factors

and discussing with experts the resulting maps) may be also useful to complement the assessment of model accuracy and to develop interpretable and parsimonious models for global soil moisture mapping. Our results suggest that a parsimonious model based on topography shows comparable accuracy with more complex model including bioclimatic and soil type classes (Figs. 6 to 8, Appendix C) and similar temporal variability (Fig. 9).

Although ML approaches generally benefit from using multiple prediction factors to represent patterns, we advocate for simpler models. The parsimonious approach (based on topography) does not necessarily reduce prediction capacity when compared with a more complex model adding bioclimatic and soil type classes and both datasets show a similar trend of soil moisture levels over time.

Our trend detection analysis reveals changes of soil moisture over time at the global scale; across areas with limited information in the ESA-CCI dataset or areas where the augmented ISMN does not exist. We observe consistent soil moisture decline at the global scale using both the soil moisture predictions based on topography and the predictions based on topography, bioclimatic features and soil classes. The soil moisture trend of the

augmented ISMN dataset was also negative (Fig. 9). These soil moisture trends bring potential implications in the calibration of future projections of the water cycle, in identifying regions of strong land–atmosphere coupling (Lorenz et al., 2015), and in quantifying the contribution of soil moisture for land-surface models (Singh et al., 2015). The negative soil





moisture trends found in this study (Fig. 10) are consistent with recent soil moisture

monitoring efforts (Albergel et al., 2013; Gu et al., 2019a). It has been shown that soil

moisture decline can be intensified by land warming (Samaniego et al., 2018), land use

change (Chen, et al., 2016; Garg et al., 2019), agricultural practices (Bradford et al., 2017), or

transformations to vegetation cover that directly affect primary productivity,

evapotranspiration rates and drought (Stocker et al, 2019; Martens et al., 2018). Furthermore,

contiguous information of soil moisture trends is increasingly needed for quantifying the

consequences of soil moisture decline in ecosystems processes such as soil respiration (Bond-

Lamberty and Thomson. 2018). Our results complement the ESA-CCI soil moisture dataset

as they identify soil moisture decline across the Congo region or the Amazon basin (Fig. 10).

These results are consistent with previous studies that have identified soil moisture decline

across the Congo region associated with reduction of precipitation rates (Nogherotto et al.,

2013), and across the Amazon basin where climate signals on plant productivity can be due

changes in soil moisture conditions (Wagner et al., 2017). Further studies are needed to fully

interpret the influence of surface or deeper soil moisture on ecological processes (Morton et

al., 2014), but we argue that surface soil moisture trends are critical to identify potentially

vulnerable regions across the world. Our examples of surface soil moisture predictions across

tropical areas (using the available ESA-CCI information) or across specific countries with

heterogeneous environmental gradients (e.g., Fig. 5), suggest that our predictions follow

expected environmental gradients and the range of values observed by the augmented ISMN,

and they show also show clear correlation with in situ precipitation records (Fig. 3).

Limitations of our approach include a) the propagation of measurement errors of the

ESA-CCI dataset used to train the k-KNN algorithm, b) the propagation of measurement

errors (and quality) of the digital elevation dataset used for calculating terrain parameters and

c) by the prediction ertherors of  k-KNN algorithm (e.g., random errors, systematic errors,

spatially autocorrelated errors). It is known that satellite-derived soil moisture estimates fail



to measure extremely dry or extremely wet conditions (McColl et al., 2017; Liu et al., 2019);
consequently, this lack of information influences the prediction capacity of our downscaling
framework and there is a need to improve modeling and measurements of these extremes. In
addition, the quality of the prediction factors will impact the quality of final prediction
outcomes. Thus, the prediction algorithm will not be able in any case, to generate a perfect

model. Therefore, it is important to provide prediction variances around soil moisture
predictions that are useful to identify areas with high or low model consistency (Fig. 4c). The
variance associated with soil moisture predictions provides novel information to assess the
strength of the relationship between the covariate space (e.g., terrain parameters, bioclimatic
and/or soil type features) and predicted soil moisture. Consequently, large prediction

variances (Appendix B) remain across areas less represented in both field measurements (Fig.
1) and across extremely dry or extremely wet conditions affecting the spatial representation
of satellite soil moisture datasets (Fig. 4a). Our prediction variances also provide insights for
future research efforts where alternative techniques are needed to provide information to
better constrain model predictions and to reduce prediction variances.

845         We discuss potential sources of prediction variance between soil moisture predictions
and datasets. Prediction variances are indicators of discrepancy levels between soil moisture
datasets (augmented ISMN, ESA-CCI, our predictions). Discrepancy between the augmented
ISMN and satellite-derived soil moisture or our downscaled datasets can be associated with
differences in the spatial representativeness of points measurements and grids surfaces

(Gruber et al., 2020). This scale mismatch has been previously identified when testing
different soil moisture patterns (Nicolai-Shaw et al., 2015) as field soil moisture records are
usually representative of <1 m$^3$ of soil while satellite and modeling estimates varies from
several meters to multiple kilometers. Soil moisture measurements (from satellites and in situ
measurements) across both water-limited environment and tropical areas are extremely

limited (Liu et al., 2019), a condition that increases prediction variances (and consequently





also increased model uncertainty). Thus, alternative modeling and evaluation frameworks and model evaluation statistics are required to provide more information to better interpret the spatial variability and dynamics of soil moisture global estimates (Gruber et al., 2020). To this end, we used in situ annual precipitation as a proxy to evaluate soil moisture estimates and found that our predicted soil moisture was better correlated than the original ESA-CCI dataset. This higher correlation may be useful for further analyses and evaluations including soil moisture and precipitation feedbacks (McColl et al., 2017) as precipitation decline has been associated with soil moisture decline in previous studies (Nogherotto et al., 2013).

Future work includes predicting global soil moisture patterns across finer pixel sizes (e.g., 1km or <1km) and higher temporal resolutions (e.g., monthly, daily), as it has been done at the regional to continental scales (Naz et al., 2020; Llamas et al., 2020; Guevara and Vargas, 2019). The current version of the downscaled soil moisture predictions is provided on an annual basis because is a temporal resolution useful for multiple ecological and hydrological studies related to large-scale ecological processes and climate change (Green et al., 2019). We recognize that there is an increasing need of soil moisture datasets with higher temporal resolutions to analyze the seasonal and short-term memory soil moisture effects after precipitation events (McColl et al., 2017). A spatial resolution of 15 km is still a coarse pixel size for detailed analysis of hydro-ecological patterns (e.g., at the hillslope scale), but the main focus of this study was to test the potential of digital terrain analysis for increasing the spatial resolution of the original ESA-CCI soil moisture dataset. Our decision for selecting a 15km pixel size was driven by the reproducibility or our approach by multiple groups without the need of HPC infrastructure. HPC is increasingly required for modeling soil moisture patterns with unprecedented levels of spatial resolution across continental scales (e.g., 3km grids, RMSE 0.04 to 0.06 $m^3m^{-3}$; Naz et al., 2020) that show comparable accuracy with our 15km grids (Fig. 8 a-c). Additionally, the increase of nearly 50% in spatial resolution suggests a larger range of soil moisture predicted values compared with the ESA-



CCI, possibly associated with scale dependent patterns of soil moisture (Fig. 5) which can be analyzed in future work.

In conclusion, to downscale (i.e., increase spatial resolution) coarse satellite soil moisture grids we used k-KNN to combine satellite soil moisture data with terrain parameters (as surrogates of topographic variability), bioclimatic and soil type classes. The validation of our soil moisture model predictions against multiple field data sources (Fig. 1) and multiple combinations of prediction factors support that digital terrain analysis can be used as a parsimonious approach for improving the spatial resolution of the ESA-CCI soil moisture

dataset (Appendix C). We provide a new gap-free and annual soil moisture dataset for 28 years provided across 15 km grids in an annual basis (1991-2018). Our results provide a global soil moisture benchmark to address the increasing need of soil moisture datasets with higher temporal and spatial resolution at the global scale.


## 5 Data availability

We provide a publicly available soil moisture dataset including working codes and information useful to replicate our results. We follow global validations standards for modeled soil moisture estimates (Gruber et al., 2020). We also provide explicit uncertainty estimates and user

guidance for interpreting and reproducing our results. The sources of information required to develop this study are:

- The soil moisture training dataset used in this study is available thanks to the ESA-CCI (https://www.esa-soilmoisture-cci.org/)
- The soil moisture validation dataset used in this study is available thanks to the ISMN

(https://ismn.geo.tuwien.ac.at/en/)



- The downscaled soil moisture predictions generated in this study are available here: https://doi.org/10.4211/hs.b940b704429244a99f902ff7cb30a31f (Guevara, et al., 2020)

  ◦ The soil moisture predictions are provided in rasters (n=28 per folder, 1991-2018) that can be imported to any GIS and they contain an accuracy report from the cross validation for each model/year in a *.csv file.

  ◦ We include a raster stack with 28 layers containing the prediction variances for each model year (1991-2018) derived from bootstrapping the k-KNN models.

  ◦ The prediction factors for soil moisture across 15km grids are also available in a R spatial pixels data frame; containing values for each pixel of:

    ▪ a) terrain parameters calculated in SAGA-GIS http://www.saga-gis.org/,

    ▪ b) bioclimatic classes from http://www.fao.org/nr/gaez/en/ transformed to a binary presence/absence, 1/0 code and

    ▪ c) the continuous classes (1 = 150 mm water per m of the soil unit, 2 = 125 mm, 3 = 100 mm, 4 = 75 mm, 5 = 50 mm, 6 = 15 mm, 7 = 0 mm) from the Re-gridded Harmonized World Soil Database v1.2 available here: https://daac.ornl.gov/SOILS/guides/HWSD.html.

    ▪ d) each soil moisture prediction contains a plot of top prediction factors affecting the accuracy (r and RMSE) computed after the cross validation strategy for each model year.

  ◦ In the same data repository, we provide the ISMN annual dataset that we used for validating (Fig. 1, green) our soil moisture predictions in a native R spatial object.

- The precipitation dataset used as alternative validation data (Fig. 1, blue) is available here: https://daac.ornl.gov/SOILS/guides/SRDB_V4.html.



- Additional soil moisture data from local studies (Fig. 1, red) across tropical areas is available here: https://iopscience.iop.org/article/10.1088/1748-9326/7/3/035704 and https://daac.ornl.gov/LBA/guides/CD32_Brazil_Flux_Network.html

- The R code used for a) to develop our soil moisture modeling and validation approach and b) to generate the base figures on this paper is available here:

  https://github.com/vargaslab/Global_Soil_Moisture.

- Step-by-step guidance for modeling satellite soil moisture using k-KNN and terrain parameters as prediction factors is available here: https://www.protocols.io/view/protocol-for-downscaling-satellite-soil-moisture-e-6cahase


As this is paper is the result of an active line of research, we will continue updating our soil moisture predictions and our results as new input data (ESA-CCI- future versions) become available. Current version covers the period of time between 1991 and 2018 and it is based on the ESA-CCI version 4.7.


**Acknowledgements**

MG acknowledges support from a CONACyT doctoral fellowship (382790). RV and MT acknowledge support from the National Science Foundation (grant #1724843). We thank Anita Z. Schwartz from the University of Delaware for her assistance preparing the global
terrain dataset.

**Author contributions**

MG, RV and MT conceptualized the project. MG performed analysis and wrote the manuscript in collaboration with RV and MT.




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



**Appendices**

**Appendix A**

We present the maps of the spatial coordinates used in our prediction approach. We developed these maps following the recently proposed method by Møller et al., (2020). In this method, latitude and longitude across the area of interest (e.g., the entire world) are rotated along several (e.g., n=6) axes tilted at oblique angles (Fig. A1) and used as prediction factors for soil attributes (e.g., soil moisture).



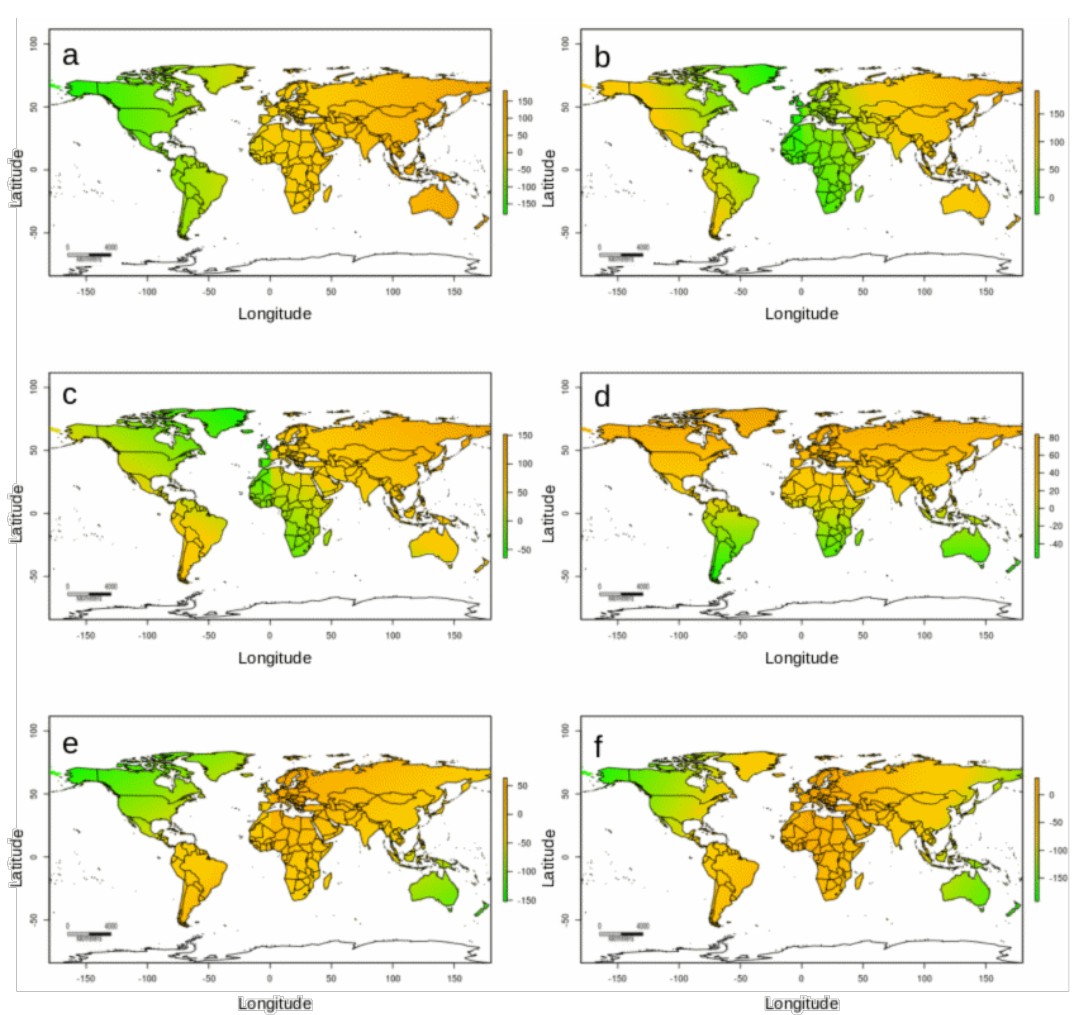


[insert] Figure A1. The variables: pi0.00 (a), pi0.17 (b), pi0.33 (c), pi0.50 (d), pi0.67 (e) and pi0.83 (f) are spatial coordinates of the global 15km grids tilted at multiple angles (n=6) used as ancillary information in order to explicitly account for the spatial structure of available soil moisture values in the geographical space.






**Appendix B**

We present the availability of data in the ESA-CCI soil moisture data for a given year (e.g., 2018) across tropical areas of the world (Figure A2a). Using this limited information only (the ESA-CCI data across the tropics) we improve the spatial representativeness of satellite soil moisture data following our prediction approach (Figure A2b). Our approach consider model uncertainty, represented by the model prediction variance after n model realizations

(Figure A2c).

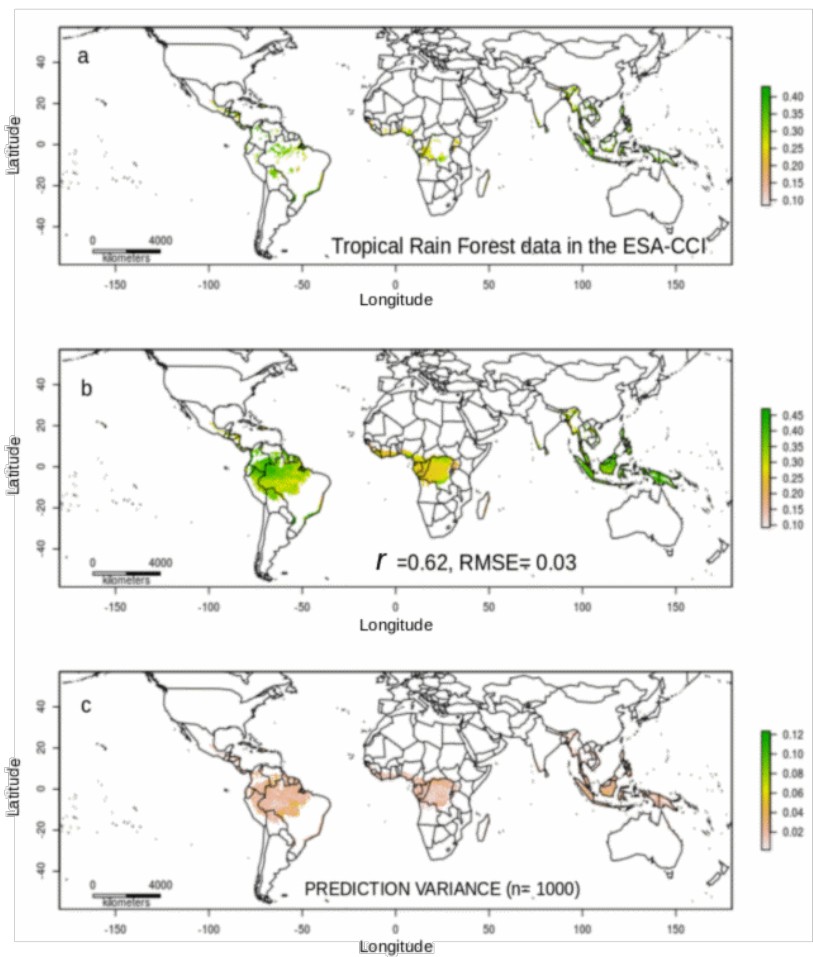

[insert] Figure A2. Soil moisture across Tropical Rain Forests of the world based on the data available in the ESA-CCI soil moisture product (4.5) for the year 2018 (a). We show the soil moisture prediction (b), the soil moisture prediction variance using only the data available for Tropical Rain Forests (c). Note that the correlation between observed and predicted decreased to 0.62, most likely due to the limited information for modeling these ecosystems, however the root mean squared error is comparable with a model using all global data (e.g., <0.04).




## Appendix C

We also present a summary of our validation of soil moisture predictions in the form of a Target Diagram (Figure A3). A Target Diagram is derived from the relation between the

unbiased RMSE, MBE (mean bias error), and RMSE. In a Cartesian coordinate system, the x-axis represents the unbiased RMSE (variance of the error), and the y-axis represents the MBE. Therefore, the distance between any point to the origin is equal to the RMSE. Because the unbiased RMSE is always positive, the left area of the coordinate system is empty with this scheme. With additional information this region may be also used: the unbiased RMSE is

multiplied by the sign of the difference between the standard deviations of model and observations. The diagram provides three different measures: whether the model overestimates or underestimates (positive or negative values of the MBE on the y-axis, respectively), whether the model standard deviation is larger or smaller than the standard deviation of the measurements (positive or negative values on the x-axis, respectively), and

the error performance as quantified by the RMSE represented as the distance to the coordinates origin (see Jolliff, et al., 2009).



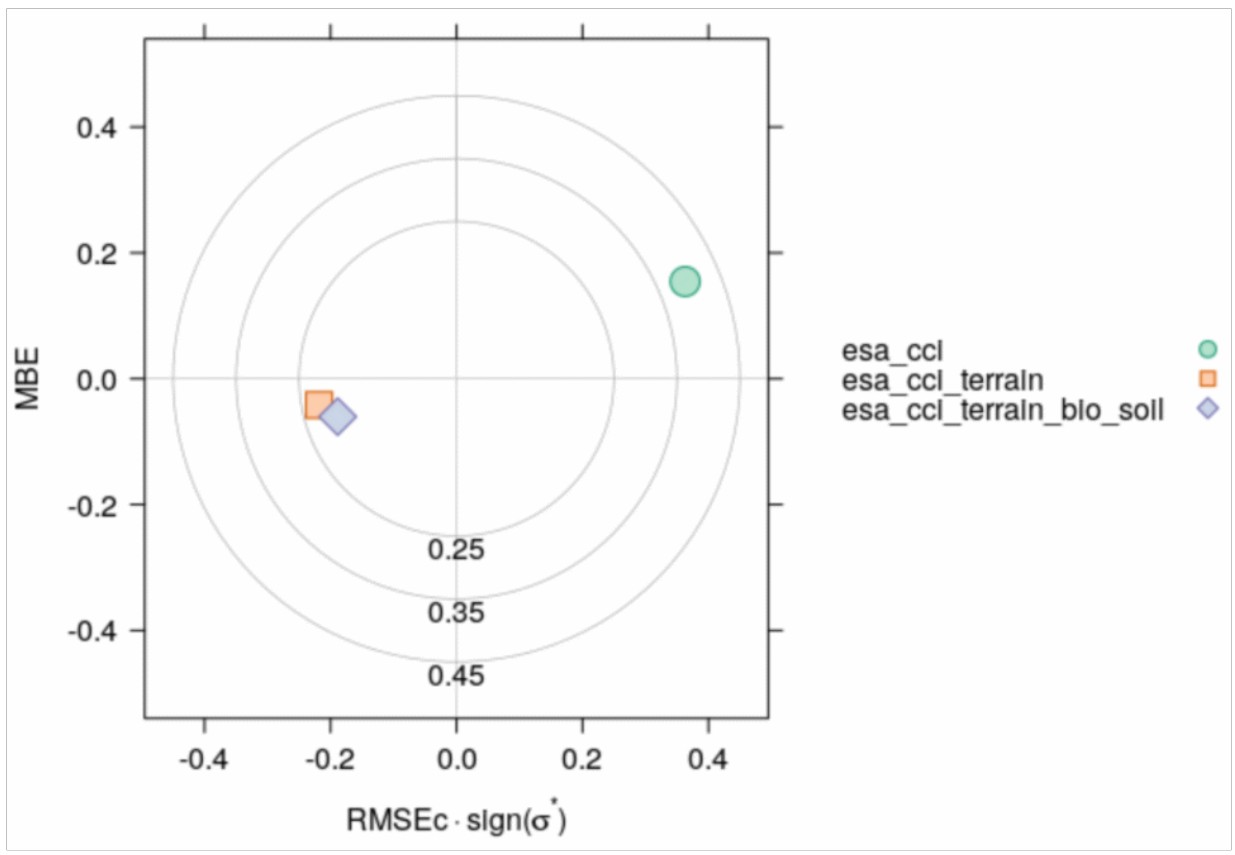

[insert] Figure A3. Target diagram showing the performance of our soil moisture predictions.
The x-axis represents the unbiased RMSE (variance units of the error), and the y-axis
represents the MBE. This figure shows that our soil moisture predictions using terrain
parameters (esa_cci_terrain) and the predictions using terrain parameters, bioclimatic and soil
type classes (esa_cci_terrain_bio_soil) show lower error levels when compared with field

data (from the ISMN) than the ESA-CCI soil moisture product (esa_cci).