# Peer review of "Gap-Free Global Annual Soil Moisture: 15km Grids for 1991-2018"

_Earth System Science Data, 2020_

## Referee Comment (RC1) · Anonymous Referee #1 · 20 Oct 2020

OVERVIEW The study presented a gap-filled global soil moisture dataset based on the ESA CCI satellite soil moisture product. The dataset is characterized by a spatial resolution of 15 km. The new filled and downscaled dataset has been validated against in situ soil moisture and precipitation data through annual comparision as well as in terms of long-term trends during the period 1991-2018. GENERAL COMMENTS The paper is well written and clear and I found the topic of interest for the readership of Earth System Science Data. However, I have some comments and doubts that should be clarify before considering the paper for publication. My comments are listed below.

1) Please, add some details about the ESA CCI SM product. Did you use the combined dataset? I think version 4.5 (or 4.9, please check out throughout the manuscript) is the latest one; 2) It is unclear to me how you have selected the data gaps in the original CCI

dataset. The ESA CCI SM product is a daily dataset, so the gaps should be present during the entire period? Did you estimate an annual mean of original CCI data? Did you find a reduction of gaps during the analysis period; 3) Bioclimatic features are not described at all; 4) It is unclear to me how the comparion with ISMN has been carried out. You extracted the original and the downscaled ESA CCI SM products over the stations locations and then? Did you estimate an annual mean of observed SM and compared to the satellite data? Are the couples drawn in Figure 8 the resulst for each year (so, 28 points)? How these couples have been estimated?; 5) The link to the downscaled product reported at line 907 is referred to the previous version of the dataset, please change it

---

## Referee Comment (RC2) · Anonymous Referee #2 · 27 Nov 2020

1. Limited usefulness of an annual average soil moisture. Some gap filled values i.e. over areas of permanent ice are not physically realistic. 2. Incorrectly referenced CCI dataset i.e. L51, 72 (as per terms&conditions https://www.esa-soilmoisture-cci.org/node/236) + manuscript does not specify which CCI product was used (passive, actice or combined). In L96: I believe the authors meant v4.7, there is no CCI SM v4.9. 3. Unnecessary level of detail in the abstract, numerical results could be saved for the end. 4. The number of used in situ ISMN records (n=13376) seems very high, nowhere does the manuscript state which sensor depths from ISMN records were selected for the validation. (As acknowledged in L65, satellited soil moisture represents 0-5cm and so only relevant in situ records should be included). 5. The overall correlation with in situ records of the original CCI dataset (L25-27 again, no

product specified) is said to be 0.3 which is much lower than the correlation values available from the official CCI product website, validation report and what a quick validation (https://qa4sm.eu/result/8098cf4a-726b-4f56-a4cb-fb180c884c5c/) all suggest (r=∼0.5). If annual mean values were validated, how were the means taken, was there a number of available observations threshold for a pixel in the CCI dataset before a mean was taken? Or could annual means be computed with at least a single observation? The same goes for in situ reference data. The selection of included in situ sensor depths would also affect the correlation metrics. 6. The 10-line Fig 2 caption repeats the preceeding text exactly; repetitive wording in other figure captions (i.e. Fig 5). 7. Section 2.2 it is unclear to me whether the model was entirely built around annualy aggregated CCI values. 8. Section 2.4: not sure if validation against a mixture of soil moisture and rainfall observations is a good approach... Later L372-373 read "The use of precipitation data for areas of the world where no in situ soil moisture validation data is supported by work of Gruber et al., (2020)." - I found no such information in the quoted paper, moreover Fig 1 shows that the used in situ rainfall stations often overlap with the soil moisture ones, contradicting that statement (unless they all don't overlap temporally). I believe 'available' is also missing from the quoted sentence. 9. The source of the precipitation records is not explained until L366 even though these data are mentioned several times before. As they are mentioned and mapped (Fig 1) together with in situ soil moisture it is easy to assume the authors refer to in situ precipitation measurements from ISMN. L162 and L368 seem to be using 'records' and 'sites' interchangeably, which is incorrect. 10. ISMN is a dynamic dataset and an access/download data would be useful. What are the 8,080 ISMN tables mentioned in L361? Confused by this number. From how many stations were the 13376 records derived? Which networks were selected and why. Again, what reference depths were included. 11. L68- there is also 1km surface soil moisture over Europe https://land.copernicus.eu/global/products/ssm

---

## Author Comment (AC1) · 22 Dec 2020

OVERVIEW The study presented a gap-filled global soil moisture dataset based on the ESA CCI satellite soil moisture product. The dataset is characterized by a spatial resolution of 15 km. The new filled and downscaled dataset has been validated against in situ soil moisture and precipitation data through annual comparison as well as in terms of long-term trends during the period 1991-2018.

GENERAL COMMENTS

The paper is well written and clear and I found the topic of interest for the readership of Earth System Science Data. However, I have some comments and doubts that should be clarified before considering the paper for publication. My comments are listed below.

[Figure]

RESPONSE: We appreciate the recognition of the importance of this dataset and the research presented. Below we address the Reviewer's comments individually.

1) Please, add some details about the ESA CCI SM product. Did you use the combined dataset? I think version 4.5 (or 4.9, please check out throughout the manuscript) is the latest one;

RESPONSE: We used the combined dataset, Version 4.5. We will clarify this issue in the revised version of our manuscript.

We recognize that the ESA-CCI soil moisture product is constantly evolving and improving. At the same time, we want to outline how our proposed downscale method (which is part of the paper contributions) is applicable across different versions of the ESA-CCI product. As a matter of fact, our method has been tested on different versions of the ESA-CCI (i.e., 4.4 and 4.5) and we have observed consistency across versions for validation at the regional (Llamas et al., 2020) and global scale (see: https://essd.copernicus.org/preprints/essd-2020-264/#discussion).

2) It is unclear to me how you have selected the data gaps in the original CCI dataset. The ESA CCI SM product is a daily dataset, so the gaps should be present during the entire period? Did you estimate an annual mean of original CCI data? Did you find a reduction of gaps during the analysis period;

RESPONSE: We use all available observations for each pixel across each year to estimate an annual mean using the original CCI v4.5 data. There are areas in the world (mainly in the tropics or deserts) with gaps that are present throughout an entire year. In the manuscript, we point out that the number of temporal and spatial gaps for a given region can be different across the several years. The years with the larger number of missing values (i.e., data not available; NAs) are between years 2003 and 2006 (Figure R1). In the revised version of our manuscript we will include more information about how gaps were quantified and how gaps vary across years (Figure R1).

[Figure]

3) Bioclimatic features are not described at all;

RESPONSE: We will include a detailed description of bioclimatic features in the methods section (e.g., 2.1 Datasets of Prediction Factors) as part of the revised manuscript.

4) It is unclear to me how the comparison with ISMN has been carried out. You extracted the original and the downscaled ESA CCI SM products over the stations locations and then? Did you estimate an annual mean of observed SM and compared to the satellite data? Are the couples drawn in Figure 8 the results for each year (so, 28 points)? How these couples have been estimated?;

RESPONSE: First we computed annual means for each year in every available location of the ISNM dataset. Then, we extracted the annual mean for each year in every corresponding location from the datasets of ESA-CCI (v4.5) and our predictions. In other words, we compare annual means for each location available in the ISNM and the corresponding locations from the datasets of ESA-CCI (v4.5) and our predictions. Consequently, these comparisons are consistent in space (i.e., locations from the ISMN and corresponding pixels in the ESA-CCI (v4.5) and our predictions) and time. We will revise the methods section (e.g., 2.4 Validation against independent in situ data) to clarify this point.

5) The link to the downscaled product reported at line 907 is referred to the previous version of the dataset, please change it.

RESPONSE: We will update the link in the revised version. The correct link is: https://www.hydroshare.org/resource/9f981ae4e68b4f529cdd7a5c9013e27e/

———————————————

[Figure]

**Fig. 1.** Figure R1. Number of data gaps or not available values (NAs) *100 in the ESA-CCI v4.5 across years during the analyzed period.

---

## Author Comment (AC2) · 22 Dec 2020

1. Limited usefulness of an annual average soil moisture. Some gap filled values i.e. over areas of permanent ice are not physically realistic.

RESPONSE: We respectfully disagree with this comment. There are multiple applications of annual soil moisture estimates that can be related with annual estimates of other variables of ecological importance. We have made our work available to the community through the Hydroshare database and we are monitoring the use of our datasets by other scientists . Our datasets (years 1991-2016 and 1991-2018) have been downloaded nearly 300 times demonstrating interest from the community. Another example of broad interest by

the scientific community is our estimates of annual soil moisture across the conterminous United States, which have been downloaded nearly 3,000 times (see:https://www.hydroshare.org/resource/b8f6eae9d89241cf8b5904033460af61/).

We believe these metrics are a solid indicator of the impact and relevance of our datasets. Examples for uses of annual average of soil moisture include topics related to soil health, trends in the carbon cycle, response of ecological communities in terrestrial ecosystems, changes in ecosystem functions, among others. We will include text in the discussion to highlight these and other potential applications. We agree with the Reviewer that values in permanent ice are not physically realistic. In the revised version of this manuscript we will not present soil moisture estimates across Antarctica or the Arctic.

2. Incorrectly referenced CCI dataset i.e. L51, 72 (as per terms&conditions https://www.esa-soilmoisturecci.org/node/236) + manuscript does not specify which CCI product was used (passive, actice or combined). In L96: I believe the authors meant v4.7, there is no CCI SM v4.9.

RESPONSE: We will revise the manuscript for consistency in data versions, and we will include the references associated with the development of the ESA-CCI according to the policy listed in the webpage. We clarify that we used ESA-CCI version 4.5.

3. Unnecessary level of detail in the abstract, numerical results could be saved for the end.

RESPONSE: We respectfully disagree with the Reviewer; we believe that quantitative results are important information in the abstract to facilitate readers in identifying the key contributions of the paper. That said, we will revise the abstract for clarity in the revised version.

4. The number of used in situ ISMN records (n=13,376) seems very high, nowhere does the manuscript state which sensor depths from ISMN records were selected for the validation. (As acknowledged in L65, satellited soil moisture represents 0-5cm and

so only relevant in situ records should be included).

RESPONSE: This number does not represent in situ monitoring stations but corresponds to the number of globally available soil moisture records from all ISMN stations between 1991 and 2018. In other words, we aggregated all available soil moisture values for each station contributing to the ISMN on a yearly basis between 1991 and 2018. We will revise the manuscript for clarity. Furthermore, we want to emphasize that the validation only with information within 0-5 cm from the ISMN is consistent. Here, we present examples of our validation approach using only information within 0-5 cm from the ISMN (Figure R1).

5. The overall correlation with in situ records of the original CCI dataset (L25-27 again, no product specified) is said to be 0.3 which is much lower than the correlation values available from the official CCI product website, validation report and what a quick validation (https://qa4sm.eu/result/8098cf4a-726b-4f56-a4cb-fb180c884c5c/) all suggest (r=âĹij0.5). If annual mean values were validated, how were the means taken, was there a number of available observations threshold for a pixel in the CCI dataset before a mean was taken? Or could annual means be computed with at least a single observation? The same goes for in situ reference data. The selection of included in situ sensor depths would also affect the correlation metrics.

RESPONSE: We will include in the revised discussions potential issues for this discrepancy. For example: the correlation presented in the ESA-CCI website represents a calculation using a different time period (between 1978 and 2020). In this study we only use data between 1991 and 2018, and we perform our comparison using the ESA-CCI on a yearly basis. We did not consider a minimum number of observations threshold for a pixel in the CCI dataset or the in situ reference data (i.e., the ISMN) before calculating global yearly means. More information on gaps (e.g., pixels with only one observation during each year) will be provided in the revised version of our manuscript.

6. The 10-line Fig 2 caption repeats the preceeding text exactly; repetitive wording in other figure captions (i.e. Fig 5).

RESPONSE: We will revise the text and remove repetitive wording.

7. Section 2.2 it is unclear to me whether the model was entirely built around annually aggregated CCI values.

RESPONSE: We built a model for each year using all available data for that specific year aggregated on an annual basis. We will revise the methods section to clarify this point.

8. Section 2.4: not sure if validation against a mixture of soil moisture and rainfall observations is a good approach... Later L372-373 read "The use of precipitation data for areas of the world where no in situ soil moisture validation data is supported by work of Gruber et al., (2020)." - I found no such information in the quoted paper, moreover Fig 1 shows that the used in situ rainfall stations often overlap with the soil moisture ones, contradicting that statement (unless they all don't overlap temporally). I believe 'available' is also missing from the quoted sentence.

RESPONSE: We clarify that our results represent independent validations: one only with soil moisture data and a second one only with precipitation data. We will revise the methods section to clarify this point. We will also revise the text in lines 372-374 following the suggestion of the reviewer.

Gruber et al (2020) provide a validation framework for soil moisture and recognize that in the absence of ground data, other variables related to soil moisture could be used as alternatives for a validation strategy. We assume that precipitation data is a closely related variable to soil moisture and we will clarify this assumption in the revised manuscript. We used precipitation data as a complementary approach for validation and comparison of ESA-CCI and our data products. We will make edits in the methods section and discussion section to clarify this approach.

9. The source of the precipitation records is not explained until L366 even though these data are mentioned several times before. As they are mentioned and mapped (Fig 1) together with in situ soil moisture it is easy to assume the authors refer to in situ precipitation measurements from ISMN. L162 and L368 seem to be using 'records' and 'sites' interchangeably, which is incorrect.

RESPONSE: We will improve consistency in this narrative and will clarify that precipitation data is not from the ISMN, but from another database. We specifically used data from the soil respiration database, which reports annual precipitation associated with locations of soil respiration measurements around the globe (Bond-Lamberty and Thomson, 2018). We will revise the manuscript for consistency of terms and avoid redundancy on how we refer to the datasets.

10. ISMN is a dynamic dataset and access/download data would be useful. What are the 8,080 ISMN tables mentioned in L361? Confused by this number. From how many stations were the 13376 records derived? Which networks were selected and why. Again, what reference depths were included.

RESPONSE: We will add more information about the ISMN version in the methods. The full ISMN dataset provides over 10,000 tables including climate, soil temperature, and soil moisture data (downloaded in August of 2019). We identified 8,080 tables with 13,376 soil moisture records across the green sites illustrated in Fig. 1. These records are available with high temporal resolution (e.g., hours to daily). Between 0-5 cm depth, we use information available from 987 sites (Figure R2) and provide this dataset along with the soil moisture predictions. We will clarify this in a revised version. Finally, we plan to include additional figures as supplementary material to show the availability of ISMN data using only information between 0-5 cm depth (Figure R2).

11. L68- there is also 1km surface soil moisture over Europe https://land.copernicus.eu/global/products/ssm2020

RESPONSE: We will include this reference in the introduction

[Figure]

[Figure]

**Fig. 1.** Fig R1 Validation plots: a) ISMN 0-5cm and soil moisture data from the ESA, b) ISMN 0-5cm and predicted soil moisture using terrain covariates, c) ISMN 0-5cm and predicted soil moisture using terrain,

[Figure]

**Fig. 2.** Fig. R2 Statistical distribution of A) soil moisture values at 0-5 cm depth according to each contributing network in the ISMN, B) years of available data and C) the spatial distribution of available

---

## Author Response (AR1)

Our manuscript was evaluated by two reviewers who provided insightful comments to improve the clarity of the work. Most comments from the reviewers were focused on the methods section and helped to improve the clarity of the manuscript. We have addressed all comments and we believe that our manuscript has substantially improved in clarity. We also revised grammar and minor edits across all the document following the comments from two reviewers. We thank Dr. David Carlson for his advice in revising a previous version of this manuscript submitted to ESSD.

**Anonymous Referee #1**

OVERVIEW The study presented a gap-filled global soil moisture dataset based on the ESA CCI satellite soil moisture product. The dataset is characterized by a spatial resolution of 15 km. The new filled and downscaled dataset has been validated against in situ soil moisture and precipitation data through annual comparison as well as in terms of long-term trends during the period 1991-2018.

GENERAL COMMENTS

The paper is well written and clear and I found the topic of interest for the readership of Earth System Science Data. However, I have some comments and doubts that should be clarified before considering the paper for publication. My comments are listed below.

RESPONSE: We appreciate the recognition of the importance of this dataset and the research presented. Below we address the Reviewer's comments individually.

1) Please, add some details about the ESA CCI SM product. Did you use the combined dataset? I think version 4.5 (or 4.9, please check out throughout the manuscript) is the latest one;

RESPONSE: We used the combined dataset, Version 4.5. We clarified this issue in the revised version of our manuscript [LINES 18, LINES 213].

We recognize that the ESA-CCI soil moisture product is constantly evolving and improving. At the same time, we want to highlight that our downscale method (which is a novel contribution from this study) is applicable across different versions of the ESA-CCI product. We emphasize that our method has been tested on different versions of the ESA-CCI (i.e., 4.4 and 4.5) and we have observed consistency across versions for validation at the regional (Llamas et al., 2020) and global scale (see: https://essd.copernicus.org/preprints/essd-2020-264/#discussion).

2) It is unclear to me how you have selected the data gaps in the original CCI dataset.
The ESA CCI SM product is a daily dataset, so the gaps should be present during the entire period? Did you estimate an annual mean of original CCI data? Did you find a reduction of gaps during the analysis period;

RESPONSE: We use all available observations for each pixel across each year to estimate an annual mean using the original CCI v4.5 data. There are areas in the world (mainly in the tropics or deserts) with gaps that are present throughout an entire year. In the manuscript, we point out that the number of temporal and spatial gaps for a given region can be different across the analyzed years. The years with the largest number of missing values (i.e., data not available; NAs) are between years 2003 and 2006 (Figure R1). In the revised version of our manuscript **[LINES 215-219]** we included more information about how gaps were quantified and how gaps vary across years (also see Appendix A).

[Figure]

Figure R1 (and Appendix A). Number of data gaps or not available values (NAs) *100 in the ESA-CCI v4.5 across years during the analyzed period.

3) Bioclimatic features are not described at all;

RESPONSE: We included a description of bioclimatic features in the methods section as part of the revised manuscript **[LINES 255-264]**.

4) It is unclear to me how the comparison with ISMN has been carried out. You extracted the original and the downscaled ESA CCI SM products over the stations locations and then? Did you estimate an annual mean of observed SM and compared to the satellite data? Are the couples drawn in Figure 8 the results for each year (so, 28 points)? How these couples have been estimated?;

RESPONSE: First we computed annual means for each year in every available location of the ISNM dataset. Then, we extracted the annual mean for each year in every corresponding location from the datasets of ESA-CCI (v4.5) and our predictions. In other words, we compare annual means for each location available in the ISNM and the corresponding locations from the datasets of ESA-CCI (v4.5) and our predictions. Consequently, these comparisons are consistent in space (i.e., locations from the ISMN and corresponding pixels in the ESA-CCI (v4.5) and our predictions) and time. We revised the methods section (e.g., 2.4 Validation against independent in situ data) to clarify this point **[LINES 392-399]**.

5) The link to the downscaled product reported at line 907 is referred to the previous version of the dataset, please change it.

RESPONSE: We have updated the link in the revised version **[LINE 978]**. The correct link is: https://www.hydroshare.org/resource/9f981ae4e68b4f529cdd7a5c9013e27e/

Anonymous Referee #2

1. Limited usefulness of an annual average soil moisture.

RESPONSE: We respectfully disagree with this comment. There are multiple applications of annual soil moisture estimates that can be related with annual estimates of other variables of ecological importance. We have made our work available to the community through the Hydroshare database and we are monitoring the use of our datasets by other scientists. Our datasets (years 1991-2016 and 1991-2018) have been downloaded > 300 times, demonstrating interest from the community. Another example of broad interest by the scientific community is our estimates of annual soil moisture across the conterminous United States, which have been downloaded nearly 3,000 times (see:https://www.hydroshare.org/resource/b8f6eae9d89241cf8b5904033460af61/). We believe these metrics are solid indicators of the impact and relevance of our datasets.
        Examples for uses of annual average of soil moisture include topics related to soil health, trends in the carbon cycle, response of ecological communities in terrestrial ecosystems, changes in ecosystem functions, among others. We have included text in the discussion to highlight these and other potential applications [LINES 835-840].

    2. Some gap filled values i.e. over areas of permanent ice are not physically realistic.

        We agree with the Reviewer that values in permanent ice are not physically realistic. In the revised version of this manuscript, we do not present soil moisture estimates across Antarctica or the Arctic [FIGURE 4 IN REVISDED MANUSCRIPT].

3. Incorrectly referenced CCI dataset i.e. L51, 72 (as per terms&conditions https://www.esa-soilmoisturecci.org/node/236) + manuscript does not specify which CCI product was used (passive, actice or combined). In L96: I believe the authors meant v4.7, there is no CCI SM v4.9.

    RESPONSE: We revised the manuscript for consistency in data versions, and we have included the references associated with the development of the ESA-CCI according to the policy listed in the webpage. We clarify that we used ESA-CCI version 4.5 [LINES 18, LINES 213].

4. Unnecessary level of detail in the abstract, numerical results could be saved for the end.

    RESPONSE: We respectfully disagree with the Reviewer; we believe that quantitative results are important information in the abstract to highlight key contributions of the paper.

5. The number of used in situ ISMN records (n=13,376) seems very high, nowhere does the manuscript state which sensor depths from ISMN records were selected for the validation. (As acknowledged in L65, satellited soil moisture represents 0-5cm and so only relevant in situ records should be included).

RESPONSE:  This number does not represent in situ monitoring stations but corresponds to the number of globally available soil moisture records from all ISMN stations between 1991 and 2018. In other words, we aggregated all available soil moisture values for each station (2185 stations) contributing to the ISMN on a yearly basis between 1991 and 2018 and compute their correlation with our annual soil moisture predictions.  Furthermore, we emphasize [LINES 393-399]  that the validation was  performed with a) information from all stations across all depths (2815 sites) and b) information from sites with available data within the first  0-5 cmof soil depth  [Figure 8 and Table 1 in revised version].

 6. The overall correlation with in situ records of the original CCI dataset (L25-27 again, no product specified) is said to be 0.3 which is much lower than the correlation values available from the official CCI product website, validation report and what a quick validation (https://qa4sm.eu/result/8098cf4a-726b-4f56-a4cb-fb180c884c5c/) all suggest (r=~0.5). If annual mean values were validated, how were the means taken, was there a number of available observations threshold for a pixel in the CCI dataset before a mean was taken? Or could annual means be computed with at least a single observation? The same goes for in situ reference data. The selection of included in situ sensor depths would also affect the correlation metrics.

RESPONSE: We included in the revised discussion potential issues for this discrepancy [LINES 896-915]. For example: the correlation presented in the ESA-CCI website represents a calculation using a different time period (between 1978 and 2020). In this study we only use data between 1991 and 2018, and we perform our comparison using the ESA-CCI on a yearly basis. We did not consider a minimum number of observations threshold for a pixel in the CCI dataset or the *in situ* reference data (i.e., the

ISMN) before calculating global yearly means. More information regarding gaps (e.g., pixels with only one observation during each year) is provided in the revised version of our manuscript [LINES 215-220].

7. The 10-line Fig 2 caption repeats the preceeding text exactly; repetitive wording in other figure captions (i.e. Fig 5).

RESPONSE: We revised the text and removed repetitive wording.

7. Section 2.2 it is unclear to me whether the model was entirely built around annually aggregated CCI values.

RESPONSE: We built a model for each year using all available data for that specific year aggregated on
an annual basis. We revised the methods section to clarify this point [LINES 300 301].

8. Section 2.4: not sure if validation against a mixture of soil moisture and rainfall observations is a good
approach... Later L372-373 read "The use of precipitation data for areas of the world where no in situ soil moisture validation data is supported by work of Gruber et al., (2020)." - I found no such information in the quoted paper, moreover Fig 1 shows that the used in situ rainfall stations often overlap with the soil moisture ones, contradicting that statement (unless they all don't overlap temporally). I believe 'available' is also missing from the quoted sentence.

RESPONSE: We clarify that our results represent independent validations: one only with soil moisture data and a second one only with precipitation data. We revised the methods section to clarify this point. We also revised the text in lines 400-4678 (revised version) following the suggestion of the reviewer. We have rewritten section 2.4 to address the comments from the reviewer [LINES 380-433].

Gruber et al (2020) provide a validation framework for soil moisture and recognized that in the absence of ground data, other variables related to soil moisture could be used as alternatives for a validation strategy. We assumed that precipitation data is a closely related variable to soil moisture, and we have clarified this assumption in the revised manuscript. We clarify that we used precipitation data as a complementary
approach for validation and comparison of ESA-CCI and our data products. We edited the methods section to clarify this approach [LINES 404-407].

9. The source of the precipitation records is not explained until L366 even though these data are mentioned several times before. As they are mentioned and mapped (Fig 1) together with in situ soil moisture it is easy to assume the authors refer to in situ precipitation measurements from ISMN. L162 and L368 seem to be using 'records' and 'sites' interchangeably, which is incorrect.

RESPONSE: We improved consistency in this narrative and clarify that precipitation data is not from the ISMN, but from another database [LINE 403]. We specifically used data from the soil respiration database, which reports annual precipitation associated with locations of soil respiration measurements around the globe (Bond-Lamberty and Thomson, 2018). We revised the manuscript for consistency of
terms and avoided redundancy on how we refer to the datasets.

10. ISMN is a dynamic dataset and access/download data would be useful. What are the 8,080 ISMN tables mentioned in L361? Confused by this number. From how many stations were the 13376 records derived? Which networks were selected and why. Again, what reference depths were included.

RESPONSE: We added more information about the ISMN version (e.g., downloaded date and webpage) in the data availability section [LINES 997-1003]. The full ISMN dataset provides over 10,000 tables including climate, soil temperature, and soil moisture data (downloaded in August of 2019). We identified 8,080 tables with 13,376 soil moisture records across the green sites illustrated in Fig. 1. These records are available with high temporal resolution (e.g., hours to daily). Between 0-5 cm depth, we use information available from 987 sites to validate our soil moisture predictions and provide this dataset along with the soil moisture predictions. We clarified this information in the revised version [395-400]. Finally, we included additional information to show the availability of ISMN data using only information between 0-5 cm depth (Fig 1 in revised version) and its use for validating our soil moisture predictions [Figure 8 a-c and Table 1 in revised version].

11. L68- there is also 1km surface soil moisture over Europe
https://land.copernicus.eu/global/products/ssm2020

RESPONSE: We included this reference in the introduction

---

## Author Response (AR2)

Final **comments from editor**:

**Line 495: a bit of redundant text here?**
LINE 495: REMOVE FROM TEXT: '3.2 Prediction Sensitivity for Different Datasets'
Response: Agree, we remove redundant text from this line.

**Line 572: please check numbers and units here**
LINE 572 REPLACE ($>0.04$ m$^3$m$^{-3}$) WITH ($>0.4$ m$^3$m$^{-3}$),
Response: Finger error. With our product we detect
higher maximum values ($>0.4$) compared with the maximum
values of the ESA-CCI ($<0.4$)